# LibMoE: A Library for Comprehensive Research on Mixture of Experts in Large Language Models

**Nam V. Nguyen**[*,◇]                                              *namnguyen16ai@gmail.com*
*FPT Software AI Center*

**Thong T. Doan**[*]                                               *doantienthongbku@gmail.com*
*FPT Software AI Center*

**Luong Tran**                                                           *luongtk@fpt.com*
*FPT Software AI Center*

**Van Nguyen**                                                          *vannth19@fpt.com*
*FPT Software AI Center*

**Quang Pham**[◇]                                            *hqpham.2017@phdcs.smu.edu.sg*
*Independent Researcher*

**Reviewed on OpenReview:** *https://openreview.net/forum?id=PB2ju8tq0n*

## Abstract

Mixture-of-experts (MoE) architectures have become a cornerstone for scaling up and are a key component in many recent large language models such as GPT-OSS, DeepSeek-V3, Llama-4, and Gemini-2.5. However, systematic research on MoE remains severely constrained by the prohibitive computational costs of training and evaluation, limiting access to large-scale studies for most researchers. We introduce LibMoE, a unified framework for reproducible, efficient, and extensible MoE research that supports both pretraining and sparse-upcycling regimes. Beyond unified implementations, the framework provides transparent analytical tools for probing routing and expert dynamics. Leveraging this foundation, we conduct a comprehensive analysis along three dimensions: (i) routing dynamics, covering expert selection patterns, routing stability and optimality, and how routing entropy reveals task specialization and expert diversity; (ii) the effect of lightweight initialization on load balancing, demonstrating how subtle changes in router initialization shape early expert utilization; and (iii) training regime differences, revealing how sparse upcycling and full pretraining exhibit distinct routing patterns and stability profiles. By lowering the barrier to entry and standardizing evaluation, along with our comprehensive analysis, LibMoE broadens access to MoE research and establishes a reliable benchmark to guide future innovations.

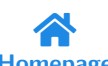 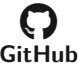

**Homepage**          **GitHub**

## 1 Introduction

Recent years have witnessed a dramatic surge in the adoption and success of deep learning models, fueled by their scalability with massive data and compute resources and their state-of-the-art performance across diverse domains. Yet, this rapid scaling has introduced significant challenges in computational efficiency. A

---

[*]Equal contribution.
[◇]Corresponding author.

prominent strategy to address these challenges is the sparse mixture of experts (SMoE) architecture, which expands model capacity without a proportional increase in computation.

First introduced by (Jacobs et al., 1991), SMoE has since achieved remarkable success across multiple fields, including large language models (LLMs) (Shazeer et al., 2017; Fedus et al., 2022; Comanici et al., 2025; Liu et al., 2024b; Jiang et al., 2024), multimodal learning (Yun et al., 2024; Lin et al., 2024a; Han et al., 2024a; Meta AI, 2025), and computer vision (Riquelme et al., 2021; liang et al., 2022; Han et al., 2024b). By activating only a sparse subset of parameters per input, SMoE substantially improves training efficiency, enabling models with hundreds of billions of parameters to be trained at manageable computational cost (Fedus et al., 2022). Interestingly, SMoE algorithms often achieved superior performances compared to their dense counterpart (Shazeer et al., 2017), which has gathered an increased interest in the community to develop more advanced algorithms and better theories to understand their behaviour (Muennighoff et al., 2024; Xue et al., 2024; Kang et al., 2025; Zoph et al., 2022; Nguyen et al., 2025b; 2024c). Consequently, SMoE has become a cornerstone methodology driving the development of next-generation intelligent systems.

However, despite rapid algorithmic advances, progress in MoE research remains fundamentally hindered by the lack of standardized frameworks. Large-scale MoE experiments are typically feasible only for groups with access to massive computational resources: training OLMoE-1B-7B, for example, required 256 H100 GPUs (Muennighoff et al., 2024), and MOMA relied on 256 A100 GPUs (Lin et al., 2024b). In contrast, the majority of researchers are confined to small-scale studies (Nielsen et al., 2025; Han et al., 2024a; Teo & Nguyen, 2024; Csordás et al., 2023; 2024; Nguyen et al., 2025a; Teo & Nguyen, 2025) or even synthetic benchmarks (Nguyen et al., 2024c;b; Yan et al., 2024), resulting in fragmented and often incomparable results. This persistent discrepancy severely undermines the potential of MoE architectures, whose true strengths emerge only in large-scale training regimes. As a result, even prominent deployed models such as Phi-3 (Abdin et al., 2024), Skywork-MoE (Wei et al., 2024), and GPT-OSS (Agarwal et al., 2025) tend to rely on conventional SMoE designs (Fedus et al., 2022), with recent advances in routing and expert selection algorithms seeing little adoption in practice (Do et al., 2023; Zhou et al., 2022; Dai et al., 2022; Wang et al., 2024b). These challenges underscore a critical need for a unified, reproducible framework that lowers the barrier to meaningful SMoE research under realistic resource constraints and enables rigorous, systematic evaluation of both established and emerging methods.

To address these challenges, we present LibMoE, a unified framework that brings together state-of-the-art SMoE algorithms, standardized training pipelines, and transparent analytical tools for rigorous investigation of routing dynamics. Our primary objectives are twofold: (1) to provide an accessible and streamlined toolkit that enables meaningful SMoE research on large language models (LLMs) under realistic resource constraints; and (2) to enable comprehensive, reproducible analysis of SMoE routing mechanisms, yielding deeper insights into expert behaviors. With LibMoE, we aim to accelerate progress and foster a more collaborative, open-source SMoE research community.

**Designing the LibMoE Framework.** To address persistent challenges in accessible and reproducible MoE research, we introduce LibMoE, a unified framework for systematic evaluation of SMoE methods across both full pretraining and sparse-upcycling regimes (Komatsuzaki et al., 2022). LibMoE accommodates a broad spectrum of model sizes and algorithmic variants, supporting rigorous comparisons from small-scale language models (0.15B and 0.68B parameters) to large-scale vision–language models (5.67B parameters). The framework integrates seven recent state-of-the-art SMoE algorithms and provides standardized pipelines for both pretraining and evaluation, enabling consistent benchmarking and cross-domain analysis. All MoE algorithm experiments are conducted under realistic resource constraints, with training times ranging from 6 to 44 hours on 4× H100 GPUs (see Table 14 for details). While our results reveal that current SMoE algorithms yield only marginal performance improvements under these settings, LibMoE substantially lowers the barrier to entry, making advanced and reproducible experimentation feasible even for research groups with limited computational resources.

**Comprehensive Analysis of Mixture-of-Experts Dynamics.** Beyond providing unified implementations, LibMoE is designed as a transparent and extensible infrastructure for systematically analyzing the behavior of SMoE models. The framework exposes modular instrumentation for monitoring routing decisions, expert utilization, load balancing, and inter-expert interactions, enabling fine-grained inspection of routing

behavior throughout both training and inference. Using this analytical capability, we conduct a large-scale, cross-domain empirical study of seven recent SMoE algorithms across multiple training regimes, including small-scale pretraining (0.15B and 0.68B parameters) and sparse upcycling at the 5.67B scale (Komatsuzaki et al., 2022). Our analysis uncovers consistent and previously under-characterized differences along three dimensions: (i) routing dynamics, reflected in expert selection stability, entropy evolution, and specialization patterns; (ii) the impact of lightweight router initialization on early-stage load balancing; and (iii) systematic divergences in routing behavior between full pretraining and sparse upcycling regimes. Together, these results demonstrate how design choices in routing and initialization influence training stability and expert utilization, and establish a reproducible analytical basis for principled comparison of SMoE algorithms.

## 2  Related Works

### 2.1  Mixture of Experts

Originally introduced by Jacobs et al. (1991), the MoE framework was first formulated as an ensemble method that combines multiple specialized models through an adaptive gating network. Early variants, including Jacobs' gating network and the Hierarchical MoE (HME) (Jordan & Jacobs, 1994), improved learning algorithms for multiclass classification (Chen et al., 1999), and achieved success in domains such as speech recognition (Gales & Airey, 2006). Eigen et al. (2013) extends the MoE to a layer in neural network, which consists of a set of experts (neural networks) and a trainable gate. These early efforts collectively shaped the theoretical and empirical foundation of MoE, setting the stage for the large-scale sparse MoE architectures that gained prominence after 2017. In 2017, Shazeer et al. (2017) introduced the SMoE model, which incorporated MoE layers into long short-term memory (LSTM) networks to dramatically scale model capacity without increasing computational cost. Building on this idea, GShard (Lepikhin et al., 2020) extended MoE to Transformers by replacing the feed-forward network (FFN) layers in the T5 (Raffel et al., 2020) architecture with MoE layers. Switch Transformers (Fedus et al., 2022) further simplified the routing mechanism and scaled training to 1.6 trillion parameters, setting a milestone for large language models (LLMs). Since then, substantial progress has been made in advancing sparsely-gated MoE architectures for LLM development, including novel routing techniques (Chi et al., 2022; Roller et al., 2021; Zuo et al., 2021; Zhong et al., 2024b; Muqeeth et al., 2023; Wu et al., 2024a; Wang et al., 2024b; Nielsen et al., 2025; Nguyen et al., 2025c; Csordás et al., 2023), stability-oriented strategies (Zoph et al., 2022; Dai et al., 2022), fine-grained expert segmentation (Park et al., 2024; Dai et al., 2024; He, 2024), shared experts (Rajbhandari et al., 2022; Liu et al., 2024a;b; Dai et al., 2024; Team, 2024), special-purpose experts (Jin et al., 2024; Yan et al., 2025), gradient update modification (Panda et al., 2025; Yang et al., 2024b), and sparse upcycling (Komatsuzaki et al., 2022; Nakamura et al., 2025; Chen et al., 2025; Hui et al., 2024); with extensions also demonstrated in multimodal learning (Shen et al., 2023; Han et al., 2024a; Li et al., 2024b; Lin et al., 2024a; Wu et al., 2024b; Yun et al., 2024). Despite the encouraging progress, a clear discrepancy remains in current MoE research. In particular, despite a proliferation of advanced algorithms and theoretical analyses, the most capable and widely deployed LLMs (Agarwal et al., 2025; Comanici et al., 2025; Liu et al., 2024a;b; Dai et al., 2024; xAI, 2024; Yang et al., 2024a; Jiang et al., 2024; Meta AI, 2025; He et al., 2024; Abdin et al., 2024; Databricks, 2024; Wu et al., 2024a) still rely on SMoE variants closely aligned with the original formulation (Shazeer et al., 2017; Fedus et al., 2022). This indicates that our current theoretical understanding remains largely confined to academic settings and has yet to meaningfully influence real-world deployments precisely the domain where SMoE could be most impactful. We attribute this gap primarily to the high entry barrier posed by massive datasets and computational requirements, which place large-scale SMoE experimentation out of reach for most research groups. Therefore, we develop LibMoE to enable training and evaluation settings that closely reflect real-world practice, including both early-stage pretraining and late-stage sparse upcycling under limited resources. This design allows for fair, comprehensive, and large-scale benchmarking of SMoE algorithms, while making state-of-the-art methods more accessible to the broader research community.

### 2.2  Analyzing and Understanding Mixture-of-Experts Models

Since the introduction of the sparsely-gated MoE layer (Shazeer et al., 2017), research has explored both the promise of conditional capacity and the challenge of expert imbalance. This has driven the develop-

ment of stability-oriented methods (e.g., StableMoE (Dai et al., 2022)), novel routing designs (e.g., Expert-Choice (Zhou et al., 2022)), and system-level optimizations (e.g., MegaBlocks (Gale et al., 2022)). Differentiable routing mechanisms, including DSelect-k (Hazimeh et al., 2021), BASE layers (Lewis et al., 2021), and ReMoE (Wang et al., 2024b), have further increased flexibility, while AdaMoE (Zeng et al., 2024) and loss-free balancing (Wang et al., 2024a) have enhanced efficiency and adaptability. Beyond algorithmic innovation, a growing body of theoretical and empirical work has examined MoE dynamics. For example, Zhao et al. (2024) derived sparsity-aware generalization bounds, while Fan et al. (2024) systematically ablated routing granularity and expert count, revealing that token-level routers tend to capture syntactic rather than semantic patterns. Other studies have characterized router behavior (e.g., Nguyen et al. (2024a)) and highlighted linguistic specialization (e.g., Antoine et al. (2024)), collectively connecting routing design to generalization, specialization, and stability. Efforts to democratize MoE research have yielded open-source frameworks such as OpenMoE (Xue et al., 2024), OLMoE (Muennighoff et al., 2024), and FLAME-MoE (Kang et al., 2025), which provide models, training frameworks, and diagnostic tools. However, the adoption of these resources is often limited by substantial computational requirements, restricting reproducibility and practical experimentation. Moreover, most prior studies focus on behaviors specific to their own MoE variants, rather than providing systematic cross-method comparisons. In contrast, our work aims to provide a unified and comprehensive empirical analysis of leading MoE algorithms under a standardized evaluation protocol. By systematically comparing a diverse set of methods and probing the effects of key design parameters, we offer deeper insights into the factors driving performance and specialization in modern MoE architectures.

## 2.3 Mixture of Experts Toolkits

Several open-source toolkits, including FastMoE (He et al., 2021), OpenMoE+t5x (Xue et al., 2024), and Tutel (Hwang et al., 2023), support the implementation of SMoE algorithms. However, these frameworks present notable limitations for contemporary research. Tutel and OpenMoE+t5x are primarily designed for large-scale pretraining on hundreds of GPUs, restricting accessibility for groups with limited resources. FastMoE, while more lightweight, lacks support for recent LLM architectures and advanced distributed training libraries such as DeepSpeed (Lian et al., 2024). In contrast, LibMoE is specifically designed to lower these barriers: researchers can train models with as few as 1B tokens for vision–language tasks and 6B tokens for language modeling representing a 1,000× reduction in data requirements compared to OpenMoE while benefiting from support for modern LLMs and modular distributed training.

# 3 Designing LibMoE

## 3.1 Preliminary: Mixture of Experts

The standard SMoE layer (Shazeer et al., 2017) consists of a router $\mathcal{R}(\cdot, W_r)$, parameterized by $W_r$, and a set of $N$ experts $\{g(\cdot, W_{e_i})\}_{i=1}^N$, each with parameters $W_{e_i}$ for $i \in [N]$. Given an input token $\boldsymbol{x}$, the router computes an affinity score vector over all experts as: $\boldsymbol{s}_{\mathcal{R}} = \sigma\left(\text{TopK}_{-\infty}(\boldsymbol{x}^\top W_r)\right)$, where $\sigma$ is a scoring function, typically implemented as a softmax or sigmoid. The operator $\text{TopK}_{-\infty}$ retains the top-$K$ values and sets the remaining entries to negative infinity $(-\infty)$, enforcing sparsity. The SMoE output is then calculated as a weighted sum of expert outputs, modulated by the affinity scores:

$$\hat{y} = \sum_{i=1}^N \boldsymbol{s}_{\mathcal{R}}^i \cdot g(\boldsymbol{x}; W_{e_i}). \tag{1}$$

In practice, $K$ is often chosen such that $K < N$ to reduce computational cost while preserving performance.

## 3.2 Vision-Language Model: LLaVA Architecture

A key challenge in training SMoE models is obtaining a massive dataset and a large amount of compute. Thus, beyond the traditional pre-training setting, we propose to incorporate SMoE training into any existing dense LLM checkpoints via the Sparse Upcycling technique (Komatsuzaki et al., 2022), which duplicates the original model to create experts and continue training them on a downstream dataset as a normal SMoE.

Consequently, we can bypass the expensive pre-training step and evaluate SMoE algorithms with the most advanced public LLMs.

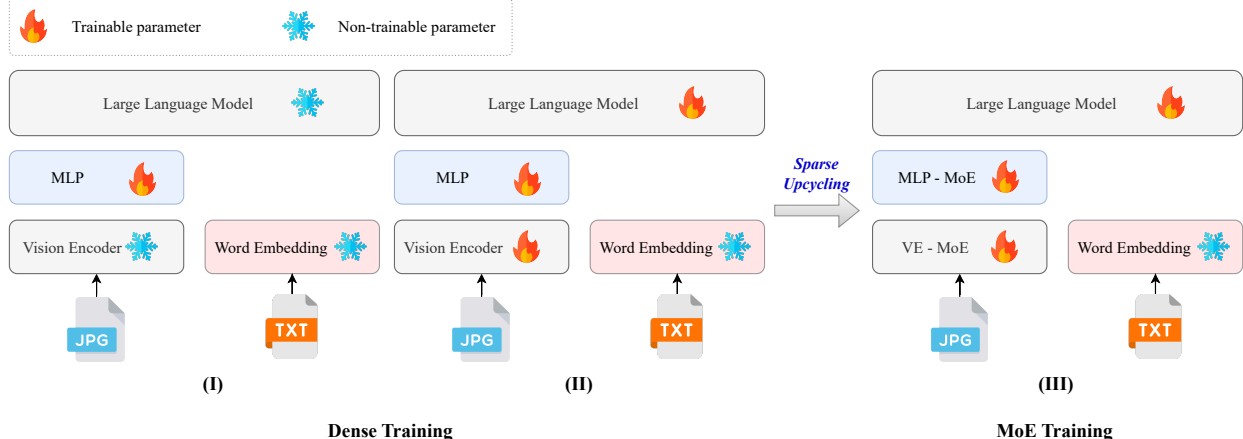

Figure 1: Overview of the LibMoE-VLM architecture and training process. In the first stage of Dense Training, only the MLP is trained to improve alignment. In the second stage, all parameters are trained. During SMoE Training, the feed-forward networks (FFNs) of the Vision Encoder (VE) and MLP Connector are used to initialize the experts within the SMoE framework, and all parameters continue to be trained.

**Training pipeline**  We adopt a vision–language pretraining task, which represents a challenging multi-modal learning setting while requiring a relatively modest amount of data to initiate training (approximately 1.4B tokens). Following the CUMO framework (Li et al., 2024b), we upcycle the LLaVA model (Liu et al., 2023a), which consists of a pretrained visual encoder, a pretrained large language model (LLM), and a randomly initialized vision–language MLP connector. Our training procedure follows a two-stage paradigm, comprising *dense training* and *sparse MoE training*. During dense training, we adopt a two-step strategy consisting of *pre-training* and *pre-finetuning*. In the pre-training stage, only the vision–language MLP connector is optimized, while both the visual encoder and the LLM remain frozen. This stage aligns visual representations with the LLM embedding space and establishes a stable multimodal interface without perturbing the pretrained backbones. Subsequently, in the pre-finetuning stage, all model parameters are unfrozen and jointly optimized using high-quality image–caption data. After dense training, we perform visual instruction tuning by converting the model into an SMoE architecture via sparse upcycling, as illustrated in Figure 1. Following prior findings (Li et al., 2024b), sparse upcycling is restricted to the MLP connector and visual encoder, since upcycling a dense LLM has been shown to be less effective than directly employing an SMoE LLM. Importantly, the dense training stage is agnostic to SMoE-specific design choices, allowing the resulting checkpoints to be reused across different SMoE algorithms. In our largest-scale experimental setting using 4×H100 GPUs, the complete training pipeline requires slightly over 54 hours in total, with the sparse MoE (visual instruction tuning) stage accounting for approximately 17 hours on the LLaVA-665K dataset or 35 hours on the OneVision (1.2M) dataset. Notably, the dense training stage (approximately 19 hours) is performed only once and reused when comparing different MoE algorithms, which significantly reduces the marginal cost of benchmarking. These results demonstrate that the proposed training setup is computationally accessible. Full specifications of model scales, datasets, and hyperparameters are provided in Appendices E.1 and F.1.

### 3.3   Pre-training LLM: Decoder-only Switch Transformer

A key design of our language modeling experiment is to establish a standardized yet flexible setup that enables fair comparison across different MoE algorithms while maintaining computational efficiency. We adopt a decoder-only Transformer architecture in the style of Switch Transformer (Fedus et al., 2022), where the feedforward sublayer is replaced by sparsely activated MoE modules. This design supports scaling from small (0.15B) to large (0.68B) models and provides a unified backbone for evaluating MoE variants. To

ensure consistency, we keep the architecture and optimization pipeline fixed across experiments, varying only the routing strategies.

**Training pipeline.** Our experimental setup builds upon the framework of Csordás et al. (2023), with adjustments tailored to our model configurations and minor refinements to the baseline hyperparameters. Our codebase supports all major forms of parallelism, including data, tensor, pipeline, and expert parallelism, ensuring scalability across model sizes. MoE variants are implemented by modularly replacing the MoE component, while keeping other training components identical. This design ensures consistency and reproducibility across experiments, allowing us to isolate the impact of routing. Full specifications of model scales, datasets, and hyperparameters are provided in Appendix E.2 and Appendix F.2.

### 3.4 Comprehensive Evaluation of SMoE Algorithms

Beyond model scaling, rigorous evaluation plays a crucial role in identifying SMoE algorithms that translate to real-world impact. Traditional machine learning studies typically follow the training–validation–testing paradigm, where separate models are trained and evaluated on individual benchmarks. Many existing SMoE works (Chi et al., 2022; Do et al., 2023) adopt this framework, training a dedicated model per benchmark.

In contrast, large language models (LLMs) have introduced a paradigm shift, wherein a single model trained on massive corpora can generalize across benchmarks without explicit fine-tuning, a setting commonly referred to as zero-shot evaluation. To align SMoE development with this emerging paradigm and real-world deployment scenarios, LibMoE is designed to support zero-shot evaluation across both language and vision-language domains.

Specifically, for vision-language modeling, we extend the LMMS-Eval framework (Zhang et al., 2024a) to evaluate the final checkpoints of various SMoE algorithms. We carefully select 12 widely used benchmarks from LMMS-Eval and report their performance, while also providing a LibMoE model loader to enable future researchers to explore nearly 100 supported benchmarks with minimal effort.

In the language modeling setting, we incorporate nine widely recognized zero-shot benchmarks, including HellaSwag (Zellers et al., 2019) and ARC-Challenge (Clark et al., 2018), which serve as canonical examples of standard evaluation datasets. These benchmarks are frequently used in the assessment of state-of-the-art models such as GPT-4 (OpenAI et al., 2023) and DeepSeek-V3 (Liu et al., 2024b), making them highly representative of real-world evaluation practices. By adopting these standardized tasks, our evaluation aligns closely with established protocols in the LLM community.

Overall, LibMoE is designed to facilitate comprehensive, zero-shot evaluation of SMoE algorithms under realistic conditions, bridging the gap between research experimentation and deployment-oriented assessment.

### 3.5 Design Principles

LibMoE is designed with a strong emphasis on modularity, extensibility, and practical usability to accelerate research on SMoE algorithms. At its core, LibMoE consists of three major modules. The MoE module provides a unified abstraction for routing strategies, supporting customizable designs such as gating mechanisms, load balancing losses, and expert selection logic. The training module handles optimization workflows across both LLM and VLM tasks, allowing researchers to integrate pretrained backbones, apply sparse up-cycling or pretraining from scratch, and utilize distributed strategies such as data parallelism and model sharding. Finally, the *evaluation module* enables seamless plug-and-play evaluation, supporting a wide range of benchmarks and task formats through standardized APIs.

In addition to providing baseline implementations of leading SMoE algorithms, LibMoE emphasizes ease of experimentation: researchers can rapidly prototype new routing designs or modify existing pipelines with minimal code changes. Its modular infrastructure allows each component routing, training, and evaluation to be extended or swapped independently. Moreover, by abstracting common functionality across tasks and model types, LibMoE lowers the engineering burden typically associated with SMoE experimentation, especially in resource-constrained settings.

## 4 Training LibMoE

### 4.1 SMoE Algorithms

To demonstrate that LibMoE is well-suited for SMoE research, we implement and benchmark seven state-of-the-art SMoE algorithms. First, we consider methods that modify or improve the router network. These include the original SMoE (Fedus et al., 2022), which uses a softmax router; the sigmoid-gated SMoE (Csordás et al., 2023), which has been shown to achieve faster convergence; and XMoE (Chi et al., 2022), which mitigates representation collapse and enhances routing capacity through input dimension reduction and cosine-normalized routing. Second, we incorporate recent designs that utilize shared expert mechanisms, such as DeepSeek-V2 (Liu et al., 2024a) with softmax routing and DeepSeek-V3 (Liu et al., 2024b) with a sigmoid router. Inspired by these designs, we implement SharedE-V2 and SharedE-V3, which adopt the shared-expert principles of DeepSeek-V2 and DeepSeek-V3, respectively. Lastly, we include MoE++(Jin et al., 2024), which augments standard SMoE with zero-computation experts and pathway-aware routing to improve efficiency and throughput, and TC-MoE(Yan et al., 2025), which introduces a ternary expert choice space and reward-based routing to reduce redundancy while enhancing model expressiveness. These diverse algorithms allow us to evaluate LibMoE's compatibility with a wide range of routing strategies and architectural innovations. A concise overview of these benchmarked SMoE variants is provided in Appendix A.

Table 1: Performance of SMoE variants with **6** versus **3** active experts on a ViT backbone (5.67B parameters). Bold values mark the best score in each column; ↓/↑ indicate that lower/higher values are preferable, respectively.

| MoE Method | AI2D | Text VQA | GQA | MM Bench | Hallusion Bench | Math Vista | MMMU | MMStar | Pope | MME | MME RW | OCR Bench | AVG Acc ↑ | AVG Rank ↓ |
|---|---|---|---|---|---|---|---|---|---|---|---|---|---|---|
| | | | | | | *LLAVA + OneVision / 1M2 samples* | | | | | | | | |
| SMoE | 69.56 | 43.93 | 61.51 | 71.31 | 46.90 | 37.90 | 41.56 | 41.23 | 86.28 | 63.33 | 27.83 | 37.50 | 52.40 | 5.96 |
| XMoE | 69.72 | 43.93 | 61.52 | 72.25 | **47.42** | 38.50 | 42.11 | **43.99** | 86.61 | 63.81 | 29.18 | **39.40** | 53.20 | 3.33 |
| $\sigma$-MoE | 69.79 | 44.69 | 61.70 | 71.74 | 47.00 | 38.40 | **43.11** | 42.08 | 86.69 | 63.80 | 29.70 | 38.40 | 53.09 | 3.33 |
| SharedE-V2 | 70.56 | **45.04** | 61.34 | 71.13 | 47.11 | **39.50** | 42.78 | 42.73 | 86.53 | 63.93 | 29.55 | 38.70 | 53.24 | 3.33 |
| SharedE-V3 | **71.92** | 44.59 | **61.93** | **72.59** | 46.37 | 38.20 | 42.33 | 43.30 | **86.78** | **64.61** | 29.29 | **39.40** | **53.44** | **2.46** |
| TC-MOE | 70.08 | 43.75 | 61.89 | 71.05 | 45.74 | 38.10 | 41.89 | 43.64 | 86.76 | 63.05 | **31.84** | 38.30 | 53.01 | 4.50 |
| MOE++ | 70.13 | 43.37 | 61.52 | 71.39 | 46.16 | 38.60 | 40.78 | 43.24 | 86.60 | 63.26 | 28.19 | 37.50 | 52.56 | 5.08 |
| | | | | | | *LLAVA / 665K samples* | | | | | | | | |
| SMoE | 65.52 | 41.51 | 61.62 | 72.25 | 41.75 | 29.50 | 41.67 | **42.24** | 87.13 | 61.05 | 32.15 | 31.70 | 50.67 | 3.42 |
| XMoE | **65.84** | 41.96 | 61.61 | 72.16 | 41.85 | **31.60** | 41.67 | 40.32 | 86.64 | 60.95 | 32.73 | **33.20** | **50.88** | **3.08** |
| $\sigma$-MoE | 65.52 | **42.08** | **61.74** | 71.22 | 40.80 | 30.30 | 41.22 | 41.99 | 86.64 | **61.44** | 32.31 | 32.50 | 50.65 | 3.71 |
| SharedE-V2 | 64.77 | 41.96 | 61.27 | 71.74 | 41.85 | 30.90 | **43.33** | 41.56 | **86.82** | 60.52 | **32.88** | 31.40 | 50.75 | 3.67 |
| SharedE-V3 | 65.58 | 42.06 | 61.26 | **72.42** | 41.43 | 30.60 | 42.44 | 41.75 | 86.81 | 60.93 | 31.47 | 32.60 | 50.86 | 3.33 |
| TC-MOE | 65.50 | 40.70 | 61.19 | 71.22 | 42.06 | 29.30 | 41.22 | 41.10 | 86.53 | 60.30 | 31.89 | 33.00 | 50.34 | 5.42 |
| MOE++ | 65.03 | 41.44 | 60.61 | 71.74 | **42.69** | 30.30 | 43.00 | 40.32 | 86.58 | 60.11 | 31.32 | 31.20 | 50.36 | 5.38 |

Table 2: Performance of SMoE variants with **66** total experts and **8** active experts in the language pre-training setting, evaluated on small-scale (0.15B) and large-scale (0.68B) models. PPL denotes perplexity for language modeling. Bold values denote the best result in each column; ↓/↑ indicate that lower/higher values are preferable, respectively.

| | MoE Method | PPL ↓ | LAM BADA | BLiMP | CBT | Hella Swag | PIQA | ARC-Easy | RACE | SIQA | Common SenseQA | AVG Acc ↑ | AVG Rank ↓ |
|---|---|---|---|---|---|---|---|---|---|---|---|---|---|
| Small Model (0.15B) | SMoE | 13.63 | 25.27 | **77.71** | 84.18 | **29.43** | 57.94 | 32.68 | 30.11 | 35.62 | 24.65 | 44.18 | 4.50 |
| | XMoE | 13.98 | 24.57 | 76.53 | 84.12 | 29.34 | 58.27 | 32.26 | 29.69 | 35.47 | 24.49 | 43.86 | 6.45 |
| | $\sigma$-MoE | 13.61 | 25.43 | 77.38 | 84.23 | 29.13 | 58.92 | 32.73 | **31.05** | 34.90 | 24.90 | 44.30 | 4.30 |
| | SharedE-V2 | 13.49 | 25.29 | 77.37 | 84.33 | 29.38 | **60.17** | **33.83** | 31.02 | 35.57 | 24.98 | **44.66** | **2.80** |
| | SharedE-V3 | **13.42** | 25.49 | 77.20 | 84.40 | 29.38 | 59.14 | 32.52 | 30.60 | 35.57 | 25.47 | 44.42 | 3.20 |
| | TC-MoE | 13.51 | **25.60** | 76.91 | 84.68 | 29.27 | 59.03 | 33.02 | 30.63 | **36.03** | **26.37** | 44.62 | 2.90 |
| | MoE++ | 13.54 | 25.45 | 77.23 | **84.83** | 29.28 | 58.49 | 33.49 | 30.11 | 35.62 | 24.49 | 44.33 | 3.85 |
| Large Model (0.68B) | SMoE | 9.51 | 37.13 | 80.47 | 89.83 | 37.49 | 64.36 | 38.22 | 33.03 | 37.41 | 26.54 | 49.39 | 5.15 |
| | XMoE | 9.66 | 35.25 | 80.38 | 89.35 | 37.19 | 64.20 | 38.99 | 32.95 | 37.77 | 28.34 | 49.38 | 5.75 |
| | $\sigma$-MoE | 9.46 | 37.56 | 81.08 | 89.57 | 37.52 | 64.91 | 39.15 | 32.68 | 37.67 | **28.50** | 49.85 | 3.60 |
| | SharedE-V2 | 9.52 | 37.11 | 80.98 | 89.93 | 37.14 | 64.36 | 38.06 | 33.17 | 36.95 | 27.35 | 49.45 | 5.25 |
| | SharedE-V3 | 9.49 | 36.88 | **81.28** | 89.65 | 37.32 | **65.72** | 38.86 | 33.12 | **38.59** | 28.09 | 49.95 | 3.60 |
| | TC-MoE | **9.38** | 37.87 | 81.21 | **90.19** | **37.95** | 64.47 | 39.28 | 33.77 | 37.92 | 27.85 | 50.06 | 2.35 |
| | MoE++ | **9.38** | **38.80** | 80.88 | 89.77 | 37.70 | 64.64 | **39.37** | **34.02** | 37.97 | 28.34 | **50.16** | **2.30** |

## 4.2 Performance Comparison

Across both vision–language modeling and language modeling settings, we observe a consistent trend: performance differences among contemporary SMoE variants remain relatively modest under matched training conditions. In the VLM sparse upcycling regime, no single method consistently outperforms others, with only marginal variations observed across benchmarks. Similarly, in language model pretraining, while recent MoE designs exhibit incremental improvements, these gains do not amount to a decisive advantage over the standard SMoE formulation. Collectively, these results reflect current practical deployments of frontier models, where the original SMoE design is often preferred for its simplicity, stability, and scalability, as opposed to more complex routing mechanisms that yield limited empirical benefits under comparable training conditions.

## 5 Analysis

Beyond empirical performance, we analyze how different MoE architectures route, specialize, and balance experts during training. We focus on expert-selection dynamics, capacity utilization, and the effects of initialization, architecture, and task domain. Our analysis spans both from-scratch pretraining (a 0.15B LLM trained on 6.55B tokens) and sparse upcycling (a VLM fine-tuned on LLaVA-665K), and additionally includes Qwen3-VL-30B-A3B Yang et al. (2025), a representative large-scale SMoE model. Together, these settings provide a unified view of routing behavior across regimes.

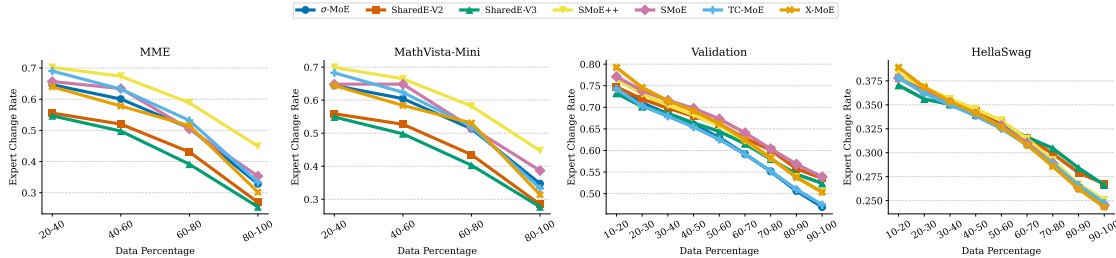

Figure 2: Impact of Training Data Percentage on Expert Selection.

**a) How Stable Is Expert Routing Throughout Training in MoE Algorithms?** This experiment investigates how training-data scale influences expert-selection dynamics across SMoE algorithms. We quantify routing stability using the *Expert Change Rate* (ECR). For a fixed evaluation set $\mathcal{D}$, we compare consecutive checkpoints and, for each token at each MoE layer, record the selected expert. We define $\mathrm{ECR}(\mathcal{D})$ as the fraction of token-to-expert assignments on $\mathcal{D}$ (aggregated across MoE layers) for which the selected expert differs between the two checkpoints. Lower ECR corresponds to more stable routing, where the router makes fewer reassignment decisions for identical inputs, whereas higher ECR indicates more volatile routing, with experts being reassigned more frequently over the course of training.

Figure 2 reports ECR throughout training for both vision-language models (MME, MathVista; sparse upcycling) and language models (Validation, HellaSwag; from-scratch pretraining). Across all settings, we observe that SMoE algorithms exhibit a consistently decreasing ECR as training progresses. Among the evaluated methods, XMoE shows stronger late-stage convergence on VLM tasks. We attribute this to its cosine-normalized routing and the smaller learning rate, which makes this setup more efficient. In contrast, during language-model pretraining, XMoE exhibits a relatively high ECR (peaking around the 40% training mark), consistent with the degraded performance reported in Table 2. We hypothesize that this degradation arises from an interaction between cosine-normalized routing and the pretraining learning rate. By bounding the router logits (up to a temperature/scale), cosine/L2 normalization can reduce the effective logit margin and make softmax gating more sensitive to the chosen scale and step size (Agarwala et al., 2020). Moreover, since normalization choices can affect gradient scaling, an overly large learning rate can exacerbate optimization instability and even trigger divergence accompanied by growing internal activations/norms (Xiong et al., 2020; Rybakov et al., 2024).

Furthermore, we observe a divergent trend in the behavior of shared expert architectures namely, SharedE-V2 and SharedE-V3 across different tasks. In vision language models (VLMs), both variants exhibit significantly higher routing stability compared to other methods, likely because $N_s$ shared experts are always active while the router only assigns tokens to $K_r$ routed experts, reducing the degrees of freedom in expert assignment relative to fully routed alternatives.

However, this advantage disappears during language model pretraining. We hypothesize that this divergence arises from the distinct initialization schemes and functional roles of shared experts. In the VLM sparse upcycling setup, SMoE layers are initialized from a pretrained dense model, where shared experts already encode general purpose visual linguistic representations. These pretrained experts act as stable anchors, enabling the router to focus its capacity on distributing tokens among non-shared experts, thereby reducing volatility in expert selection. In contrast, language model pretraining starts from random initialization. Here, both the router and shared experts must co-adapt from scratch. As the shared experts' representations evolve over

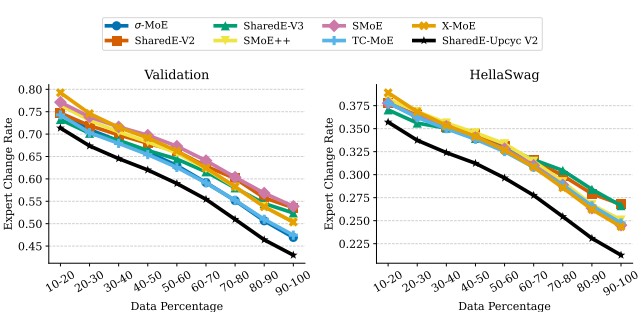

Figure 3: Effect of upcycled shared experts trained on prior tasks on routing behavior, measured by expert change rate during language model pretraining.

time, their influence on the final output changes accordingly. This compels the router to continuously revise its decisions not just for load balancing but also to adapt to the shifting behavior of shared experts in order to minimize the task loss. The result is a tight coupling between unstable shared experts and routing dynamics, which leads to higher expert change rates and delayed convergence.

To validate this hypothesis, we conduct a controlled experiment (Figure 3) by weakening the shared expert's influence: we upcycle only the shared expert from a pretrained checkpoint while randomly initializing the rest of the model, referred to as SharedE-Upcyc V2. The results show that when the shared expert is strong and stable, the router can focus on allocating inputs to task-specific experts leading to a lower expert change rate compared to other baselines. Interestingly, this stabilizing effect diminishes as training data increases. In large-scale regimes (e.g., Table 1, trained on over 1M samples), SharedE-V2 and SharedE-V3 not only preserve routing stability but also achieve top performance. This confirms that shared expert architectures are particularly beneficial in data-rich scenarios, where their pre-initialized knowledge complements the router's optimization.

Taken together, these findings highlight that routing stability is shaped by the training regime. In from-scratch language-model pretraining, both the router and experts must co-adapt under random initialization, so early-stage stability is critical to prevent prolonged assignment churn. In contrast, VLM sparse upcycling starts from a pretrained dense checkpoint, where experts can serve as stable general-purpose anchors; consequently, routing tends to evolve more smoothly and robustness becomes more important than rapid early convergence. Therefore, effective sparse MoE designs should account for both regimes: fast stabilization during from-scratch pretraining and sustained robustness during upcycling.

**b) Are the Selected Experts Truly Optimal in MoE Algorithms?**  To assess whether SMoE algorithms fully exploit their expert capacity, we conduct a diagnostic routing perturbation experiment across seven architectures. Specifically, we intentionally degrade the routing decision by replacing the top-1 expert (i.e., the expert with the highest affinity score) with the top-$(K+1)$ expert, and quantify the resulting performance change using $\Delta$Total Score (see Appendix I.1 for details). Here, a higher $\Delta$Total Score indicates improved performance relative to the original routing configuration. This controlled perturbation allows us to systematically measure each model's sensitivity to suboptimal expert assignments and to identify potential inefficiencies in expert utilization. Figure 4 reports the cumulative accuracy drop across benchmarks under two settings: replacing only the top-1 expert (*DropTop1*) and replacing both the top-1 and top-2 experts (*DropTop1&2*). As expected, most SMoE variants exhibit performance degradation, confirming that precise

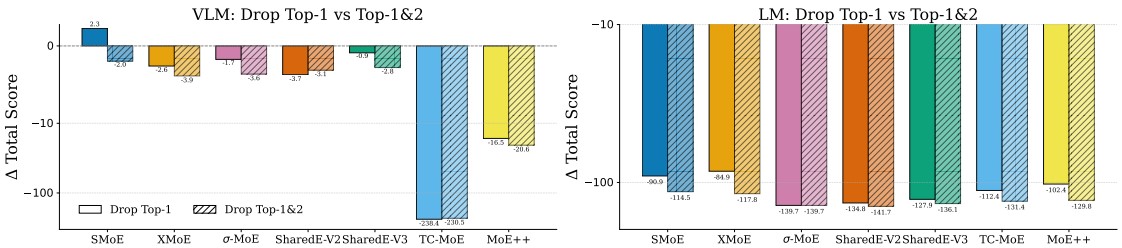

Figure 4: Performance of SMoE variants under routing perturbations, where the top-1 expert is replaced by the top-$(K+1)$ expert in vision–language and language modeling tasks.

expert selection is critical to maintaining model quality. Overall, in the language modeling (pretraining) setting, all methods suffer large and relatively uniform performance drops, reflecting a consistent reliance on precise expert routing across architectures. In contrast, vision-language models exhibit greater divergence. Notably, SMoE shows a counterintuitive **a +2.3 gain in $\Delta$Total Score** under the *DropTop1* perturbation suggesting that its original routing configuration may have failed to effectively utilize its expert capacity. This exposes a potential inefficiency in the router, where suboptimal expert choices actually yield better outcomes. Conversely, TC-MoE suffers the most severe degradation, highlighting its strong dependence on highly optimized, task-aligned routing decisions.

In summary, this experiment highlights that the impact of routing quality is fundamentally intertwined with architectural design across SMoE variants. Architectures such as TC-MoE and MoE++ exhibit pronounced performance degradation under routing perturbations, indicating that their gains rely heavily on precise and task-aligned expert selection. This sensitivity reveals a strong coupling between routing robustness and the effectiveness with which the architecture exploits its representational capacity. However, the results also demonstrate that routing robustness alone is insufficient to guarantee the highest absolute performance. Each SMoE variant is constrained by an inherent architectural capacity ceiling, beyond which improvements in routing precision yield diminishing returns. Consequently, optimal performance depends not only on selecting the "right" experts, but on whether the routing mechanism is capable of fully leveraging the architectural capacity available. These findings suggest that routing behavior should be interpreted in conjunction with architectural design, rather than as an isolated indicator of model quality.

**c) How Does Normalized Expert-Allocation Entropy Reveal Domain Specialization Across SMoE Variants?** In Figure 5, we analyze domain specialization through the normalized Expert Allocation Entropy (EAE) metric (see Appendix I.2), which measures how evenly routing decisions are distributed across experts for each MME subtask. EAE provides a direct view of the specialization–coverage trade-off: lower EAE indicates that the router repeatedly relies on a smaller subset of experts, whereas higher EAE indicates more uniform expert usage. This makes the metric particularly useful for distinguishing architectures that develop strong domain-specific specialization from those that maintain broad, balanced coverage across tasks.

Across subtasks, a consistent pattern emerges: reasoning-heavy tasks such as `text_translation`, `code_reasoning`, and `numerical_calculation` tend to yield lower EAE than simpler tasks. In other words, as the task becomes more compositional or algorithmic, the router shifts from broad expert sharing toward sharper expert separation. The same directional trend is also visible in Qwen3-VL-30B-A3B: even at larger scale, reasoning-oriented subtasks still reduce EAE, suggesting that domain-specialized routing is not merely a small-model artifact.

The LibMoE variants separate into three clear regimes. TC-MoE shows the lowest EAE and the strongest concentration, with normalized EAE dropping to about 0.64 in the most concentrated cases. This indicates aggressive specialization, but it also suggests *over-specialization*: much of the expert pool is used rarely, which can reduce coverage and make performance more sensitive to routing errors. This interpretation is consistent with the strong degradation of TC-MoE under routing perturbation in Figure 4. XMoE occupies a more favorable middle ground. Relative to the more uniform baselines, it maintains lower entropy, but it also exhibits the largest task-wise spread in EAE, ranging from about 0.78 to 0.85. This wider range

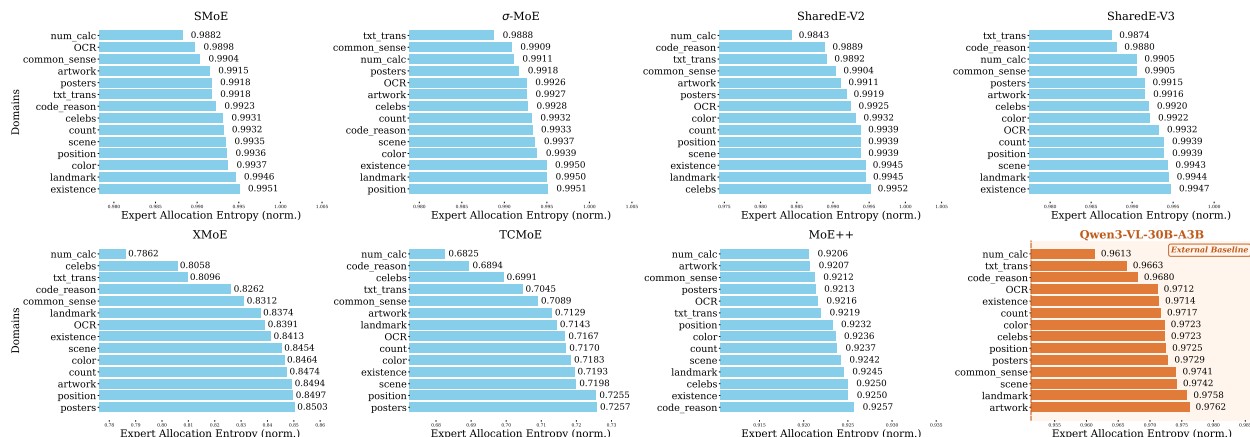

Figure 5: Normalized expert-allocation entropy across domains for seven LibMoE variants, with Qwen3-VL-30B-A3B included as a large-scale baseline.

suggests task-adaptive specialization: XMoE concentrates routing more aggressively for difficult reasoning tasks, yet distributes computation more broadly on simpler tasks, avoiding permanent collapse onto a small expert subset.

By contrast, SMoE, $\sigma$-MoE, MoE++, SharedE-V2, and SharedE-V3 show highly stable EAE curves, with variation below one percentage point across subtasks. These routers therefore prioritize consistent expert coverage over strong task-specific separation. Such behavior may sacrifice some fine-grained specialization, but it can also improve robustness by preventing a small number of experts from dominating the routing decisions. More importantly, it suggests that any task sensitivity in these architectures may come less from *which* experts are selected and more from *how* the selected experts are weighted—a question we analyze next.

Overall, EAE shows that the most effective routing behavior is not simply the lowest-entropy one. Extremely low EAE can reflect useful specialization, but it can also signal brittle expert collapse; extremely flat EAE improves balance, but may under-express domain structure. The most compelling behavior is task-adaptive entropy: specialize when the input demands it, while preserving enough coverage to remain robust. This clarifies the main message of Figure 5: TC-MoE tends to over-specialize, the balanced family tends to under-specialize, and XMoE comes closest to navigating the middle ground, while Qwen3-VL-30B-A3B shows that the same trade-off remains relevant even at larger scale.

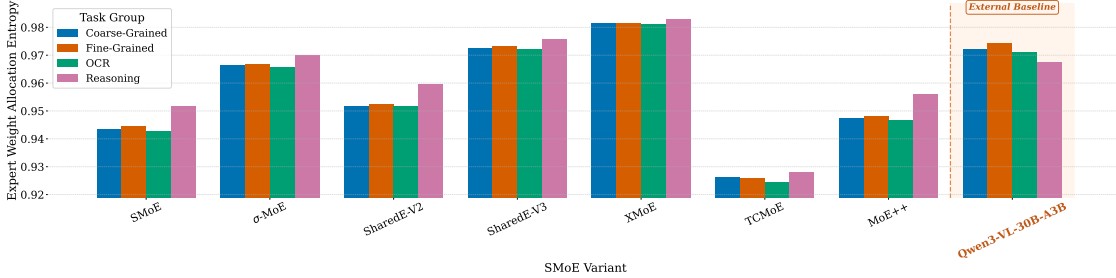

Figure 6: Normalized entropy of expert weight allocation across different routing strategies. Higher values indicate more uniform sharing of router weight among the selected experts.

**d) How Do Sparse MoE Methods Allocate Weights Across Experts and Tasks?** Beyond analyzing how expert selection varies across subtasks, we also investigate how the router distributes the weight mass of $s_{\mathcal{R}}$ among the selected experts, a critical factor that influences the model's output behavior.

In Figure 6, we report the normalized entropy of expert-weight allocation (EWA), computed as the Shannon entropy of the softmax router weights over the selected experts, normalized by the maximum entropy of a $K$-dimensional vector, and averaged over samples within each task group (Coarse-Grained, Fine-Grained, OCR, and Reasoning). Higher EWA indicates that the router distributes weight more evenly across the selected experts, whereas lower EWA indicates that it places most of the mass on only a few of them. A first notable result is that most methods achieve relatively high EWA, showing that their routers generally distribute weight broadly rather than allowing a single selected expert to dominate. In other words, once the top-$K$ experts are chosen, these methods tend to let multiple experts contribute meaningfully to the final output. This interpretation is further reinforced by the external baseline Qwen3-VL-30B-A3B, which also maintains consistently high EWA across all four task groups. At the same time, for nearly every architecture, the EWA values remain relatively stable across Coarse-Grained, Fine-Grained, OCR, and Reasoning tasks, suggesting that weight-allocation style is largely an intrinsic property of the routing mechanism rather than a strongly task-dependent effect.

The methods themselves separate into clear regimes. At the high-entropy end, XMoE shows the most uniform allocation, with $\sigma$-MoE and SharedE-V3 also exhibiting strongly balanced behavior. In these models, several selected experts tend to contribute with comparable weight, which suggests fuller use of the available expert capacity. At the opposite extreme, TC-MoE records the lowest EWA across all task groups, again with very little inter-task variation. This indicates a consistently sharper allocation pattern in which the router relies heavily on a small subset of experts regardless of task type. Such concentrated routing may encourage stronger specialization, but it also reduces expert diversity, leaves part of the model capacity underutilized, and makes the system more vulnerable when the dominant experts are not well matched to the input. SMoE, MoE++, and SharedE-V2 occupy an intermediate regime, balancing concentration and sharing more evenly.

Overall, the EWA analysis shows that expert utilization depends not only on *which* experts are selected, but also on *how* the router allocates weight among them. High-EWA methods behave more like collaborative routers, allowing multiple experts to participate meaningfully in each decision, whereas low-EWA methods behave more like decisive routers that place most of the computation on a narrow subset of experts. This provides a complementary lens for interpreting performance differences across SMoE variants: the central trade-off is not simply specialization versus balance, but how effectively each architecture converts its selected experts into useful computation.

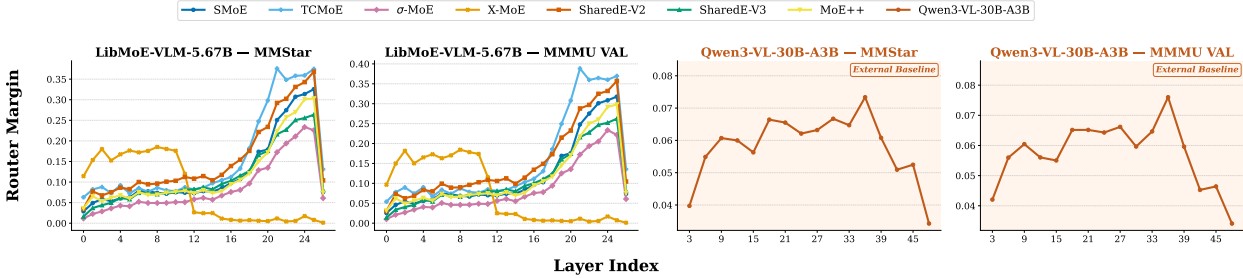

Figure 7: Layer-wise router margin in the vision–language setting, computed as the gap between the top-1 and top-2 routing scores. Left: LibMoE-VLM-5.67B evaluated on MMStar and MMMU Validation. Right: Qwen3-VL-30B evaluated on the same benchmarks.

**e) Does Router Confidence Grow with Depth in MoE Networks?** Router margin measures how strongly the router prefers the top-1 expert over the runner-up, defined as the difference between the top-1 and top-2 routing scores (see Appendix I.3). Figure 7 reveals a consistent depth-wise pattern rather than a simple monotonic increase. For most LibMoE-VLM methods on both MMStar and MMMU Validation, margins remain modest through the early and middle layers, rise clearly in the later layers, and then drop at the final layer. Importantly, this is the same qualitative trend observed in the large-scale Qwen3-VL-30B reference model: router confidence strengthens with depth before relaxing at the last layers. This alignment is notable because Qwen3-VL-30B has substantially more layers, yet LibMoE still reproduces the

same large-scale routing trajectory, suggesting that its routing dynamics capture a meaningful depth-wise behavior rather than an artifact of model size.

Across methods, TC-MoE reaches the highest late-layer margins, indicating the sharpest expert preference near the end of the network. SMoE, SharedE-V2, $\sigma$-MoE, SharedE-V3, and MoE++ all broadly follow the same late-rise-then-final-drop pattern, differing mainly in magnitude. XMoE is the main exception: it begins with relatively large margins in shallow layers, then collapses to near-zero after roughly the first half of the network, indicating increasingly weak separation between its top two experts in deeper layers. Qwen3-VL-30B provides the large-scale reference here: although its trajectory is smoother and spread over many more layers, it still exhibits the same overall increase toward later layers followed by a downturn at the end. Overall, the key observation is not merely that some LibMoE variants become more decisive with depth, but that several of them mirror the qualitative depth profile of Qwen3-VL-30B despite operating at a much smaller scale.

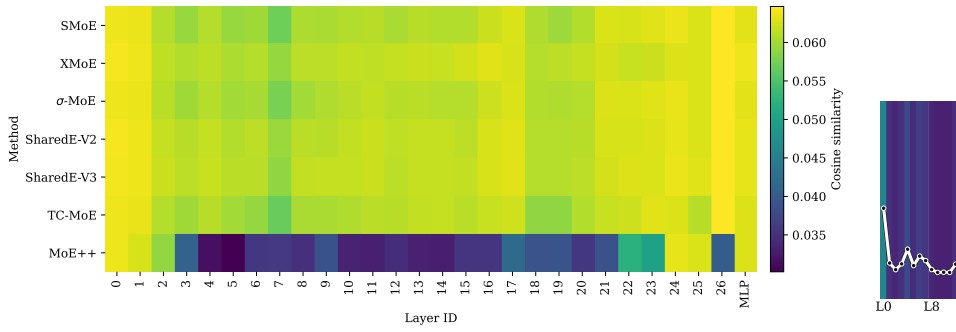
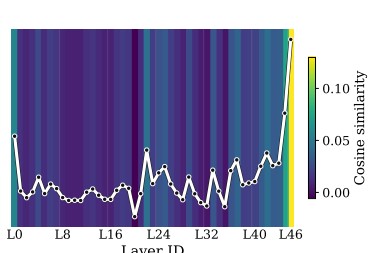

(a) Expert similarity across LibMoE-VLM methods under sparse upcycling.

(b) Expert similarity analysis for Qwen3-VL.

Figure 8: Expert similarity under sparse upcycling. Left: layer-wise expert similarity across LibMoE-VLM methods; right: expert similarity profile for Qwen3-VL on MMStar.

**f) Do Experts Exhibit Similarity in Sparse Upcycling?**   In sparse upcycling, experts are initialized by duplicating dense MLP layers, raising the question of whether they remain similar or progressively specialize during training. To examine this, Figure 8a reports the layer-wise cosine similarity of expert output weights, where values close to zero indicate greater divergence between expert outputs, while Figure 8b provides a corresponding profile for Qwen3-VL on MMStar.

We find that most MoE variants exhibit low similarity, indicating that experts diverge and specialize over time despite identical initialization. Similarity is consistently higher at the input and output layers than in intermediate layers, suggesting that boundary layers retain more shared structure while middle layers undergo stronger specialization. This pattern is also clearly observed in Qwen3-VL, whose layer-wise profile shows the same trend of elevated similarity near the input and output layers with reduced similarity in the middle. The effect is most pronounced in MoE++, which employs both copy- and zero-experts and thus encourages greater diversity in intermediate layers. Overall, excessive expert similarity does not persist, supporting the validity of comparing MoE methods under the sparse upcycling regime.

**g) How efficient is it to initialize a weight router network for load balancing?**   Prior works have explored various load-balancing techniques to mitigate the expert imbalance problem in MoE models (Fedus et al., 2022; Wei et al., 2024; Wang et al., 2024a). However, aggressive balancing may lead to routing collapse (Shazeer et al., 2017). Recent studies, such as DeepSeekV3 (Liu et al., 2024b), argue that overly strong load-balancing loss can degrade model performance. To address this, they employed auxiliary-loss-free strategies (Wang et al., 2024a) and reduced the coefficient of the standard balancing loss (Fedus et al., 2022). However, these approaches require introducing additional parameters.

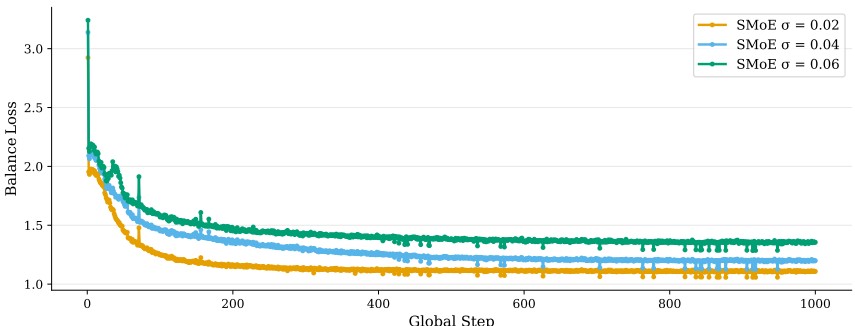

Figure 9: Impact of initialization standard deviation on balance loss dynamics in SMoE models.

Interestingly, in our study, we observe that the initialization standard deviation (std) of the router network weights alone can influence balance dynamics even without modifying the loss function. As shown in Figure 9, we initialize the SMoE router network with different std values (0.02, 0.04, and 0.06), and train for 1000 steps using the standard balance loss (Fedus et al., 2022). The results clearly show that a smaller initialization std leads to better load balance under the same softmax routing configuration. This suggests that subtle changes in the initial logit distribution can enhance routing diversity, offering an alternative or complementary axis to tune alongside both the auxiliary loss coefficient and the initialization std.

## 6    Summary of Key Results

Across both vision–language sparse upcycling and language-model pretraining, LibMoE shows that performance differences among current SMoE algorithms remain modest once compute and data budgets are matched (Tables 1 and 2). The multi-seed results in Appendix C reinforce this conclusion: benchmark-level rankings may shift slightly, but no method emerges as a consistent winner across regimes. More sophisticated routing mechanisms do not reliably deliver large gains over vanilla SMoE at the scales we study, suggesting that architectural complexity alone is rarely the decisive factor. At the same time, the value of LibMoE lies well beyond benchmark ranking. Our analyses provide a broader and deeper view of MoE routing across architectures, regimes, and scales, revealing how experts stabilize, specialize, allocate weight, and respond to perturbations during training. That said, SharedE-V2 and SharedE-V3 stand out as strong practical choices because they remain competitive in quality while offering better peak memory usage, training efficiency, and inference time than most alternatives (Appendix J). The main contribution of this section, therefore, is not only to distill practical principles, but also to show how analytical evidence can clarify how MoE architectures should be chosen, configured, and improved.

**Principle 1: Prioritize methods that remain strong across training regimes.**   A central practical result of LibMoE is that the same small set of methods remains competitive across both vision–language sparse upcycling and from-scratch language-model pretraining, even when their routing dynamics differ. SharedE-V2 and SharedE-V3 are the clearest examples. In vision–language sparse upcycling, they exhibit the most stable routing and among the strongest results in our higher-data VLM setting (Figures 2 and 3, Table 1). In language-model pretraining, their routing is more volatile early in training, yet they still finish among the top-performing methods in our LM benchmark. The main takeaway, therefore, is not that the two regimes favor entirely different methods, but that strong methods can remain strong across regimes while arriving there through different optimization trajectories.

The regime-specific analyses remain important because they explain *why* these cross-regime winners behave differently during training. In sparse upcycling, pretrained shared experts likely provide stable anchors that allow the router to focus on assigning residual capacity to task-specific experts. When this anchor is weakened in SharedE-Upcyc V2, the stabilizing effect diminishes, supporting this interpretation. In pretraining, by contrast, the router and shared experts must co-adapt from random initialization, creating an early optimization bottleneck rather than a fundamental weakness of the method. This makes shared-expert

approaches promising targets for future work on initialization, routing stabilization, and training strategies that better unlock their eventual capacity. A similar pattern appears in XMoE: it remains competitive across settings, but its cosine-normalized routing is more stable in sparse upcycling with smaller learning rates than in pretraining, where sensitivity to scale interacts less favorably with optimization. The broader lesson is therefore twofold: methods that perform well in both regimes deserve priority as practical defaults, and evaluating both regimes is still necessary because similar end performance can mask meaningfully different routing dynamics and optimization bottlenecks.

**Principle 2: Specialization and robustness trade off; sharper routing is not always better.** Our entropy, weight-allocation, router-margin, and routing-perturbation analyses together place the seven algorithms on a spectrum from collaborative to highly decisive routing (Figures 4, 5, 6, and 7). TC-MoE lies at the decisive extreme: it concentrates routing mass on a small subset of experts, reaches the sharpest late-layer margins, and exhibits strong specialization. However, it also suffers the largest performance drops under *DropTop1* and *DropTop1&2*, indicating that sharp routing can be brittle when the preferred experts are unavailable. At the other end, the SMoE family maintains much more uniform allocation, a pattern that is also broadly consistent with the large-scale Qwen3-VL-30B-A3B reference. Importantly, vanilla SMoE even improves under *DropTop1* in the VLM setting, suggesting that "top-1" routing is not always the most effective use of the available experts. XMoE often occupies a useful middle ground, adapting its degree of specialization across subtasks rather than pushing toward uniformly sharp routing. The resulting lesson is that new routing mechanisms should be judged not by sharpness alone, but by whether they achieve task-adaptive specialization without sacrificing robustness.

**Principle 3: Prefer lightweight interventions before adding architectural complexity.** One of the most practically useful findings in LibMoE is that reducing the router-weight initialization standard deviation from 0.06 to 0.02 materially improves early load balancing, without changing the loss or architecture (Figure 9). Combined with the narrow performance gap in Tables 1 and 2, this suggests that simple configuration choices can yield gains comparable to those of more elaborate routing changes in our setting. In other words, before proposing a new MoE architecture, it is worth exhausting cheaper levers such as initialization scale, learning-rate interaction with routing normalization, and the active-expert budget. A more complex method should therefore justify itself by delivering clear gains in accuracy, robustness, or effective capacity utilization rather than marginally refining the router in isolation.

Taken together, these three principles convert LibMoE from a benchmark suite into a decision framework. Practitioners can use them to prioritize methods that remain strong across training regimes, to avoid equating sharper routing with better routing, and to exhaust low-cost configuration improvements before adding architectural complexity. Regime-specific analyses still matter, but primarily as diagnostic tools: they help explain why a method succeeds, where its optimization bottlenecks lie, and which interventions are most likely to improve it. In this sense, LibMoE enables researchers to systematically evaluate MoE algorithms across both early-stage regimes, where from-scratch pretraining is unstable and routing is still co-adapting with the experts, and late-stage regimes, where sparse upcycling starts from a stronger pretrained foundation. This dual-regime view offers actionable insights that go beyond standard performance metrics, connecting final accuracy to routing stability, expert specialization, robustness under perturbation, and effective capacity use. For future algorithmic work, these principles also define a stronger evaluation bar: a new method should remain competitive with vanilla SMoE under matched compute across regimes, stay robust under routing perturbation, and demonstrate either more appropriate task-dependent specialization or better use of model capacity. Finally, we emphasize that these conclusions are directly supported at the accessible scales studied in LibMoE. At the same time, the strong correspondence between the behavioral conclusions we observe in LibMoE and those seen in Qwen3-VL-30B-A3B at much larger scale and trillion-token training suggests that LibMoE captures routing phenomena that remain relevant in practice. This, in turn, supports LibMoE as a reliable framework for researchers to explore and evaluate the next generation of MoE methods.

# 7 Conclusion

In this work, we presented LibMoE, a unified and extensible framework that enables reproducible research on Mixture-of-Experts models across language and vision–language domains. By integrating seven recent algorithms within standardized pipelines and offering analytical tools for routing, specialization, and load balancing, LibMoE lowers the barrier for experimentation under resource constraints while providing fair and systematic comparisons. Particularly, all experiments in this study were conducted on a cluster of 4×H100 GPUs, with the longest experiment taking just under 54 hours for VLMs (of which only 35 hours were spent on MoE training) and 43 hours for language model pretraining. Our comprehensive empirical study reveals that no single method universally dominates, with performance and stability shaped by task type, initialization, and expert design, underscoring the need for flexible and transparent evaluation. We hope that by bridging the gap between algorithmic innovation and practical deployment, LibMoE will not only accelerate MoE research but also provide the community with a principled foundation for building the next generation of scalable, efficient, and interpretable large models.

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

# Supplement to "LibMoE: A Library for Comprehensive Research on Mixture of Experts in Large Language Models"

## A  Overview of Benchmarked SMoE Algorithms

LibMoE benchmarks seven representative sparse Mixture-of-Experts (SMoE) algorithms that cover the main design axes explored in the literature: the router scoring function, the expert-selection rule, and structural modifications such as shared experts or zero-computation experts. To make the benchmark easier to follow, we briefly summarize the role of each method below.

**Standard SMoE.**  We use the standard sparsely gated MoE layer introduced by Shazeer et al. (2017) and later simplified for Transformer language models by Switch Transformer (Fedus et al., 2022) as our canonical baseline. In this formulation, a conventional dense feed-forward block is replaced by $N$ parallel FFN experts and a lightweight token-level router. For an input token $x$, the router computes expert logits with a linear projection, converts them to routing weights with a softmax, selects the top-$K$ experts in our benchmark configuration, and combines their outputs through the selected routing weights. This design is appealing because it is simple, computationally efficient, and widely adopted in practice.

**XMoE.**  XMoE is motivated by the representation-collapse analysis of Chi et al. (2022), which shows that standard sparse MoE routing can push token hidden states toward expert embeddings, causing routed representations to cluster around experts and reducing representation diversity. To alleviate this issue, XMoE changes the routing score from an unnormalized dot product to a cosine-style affinity: token representations are first projected into a lower-dimensional routing space, and both projected token representations and expert embeddings are L2-normalized before computing routing scores. As a result, expert selection depends primarily on angular similarity rather than vector magnitude, which helps reduce the collapse tendency in the routing representation space. In LibMoE, XMoE provides a useful contrast to conventional dot-product-based sparse routing.

**$\sigma$-MoE.**  $\sigma$-MoE replaces softmax competition with sigmoid-based expert scoring, following the sigmoid-gating formulation studied in sparse MoE language models (Csordás et al., 2023). Unlike softmax routing, where selected experts compete through a normalized probability distribution, sigmoid routing scores each expert more independently before the selected expert outputs are combined. This makes $\sigma$-MoE a useful reference for isolating how the router scoring function affects routing dynamics, load balancing, and expert utilization.

**SharedE-V2 and SharedE-V3.**  These two variants are inspired by the shared-expert design used in DeepSeekMoE and its later DeepSeek-V2/V3 instantiations (Dai et al., 2024; Liu et al., 2024a;b). In this design, part of the expert capacity is assigned to shared experts that are activated for every token, while the remaining routed experts are selected conditionally by the router. This creates an always-on pathway for general-purpose computation alongside token-dependent sparse computation. In LibMoE, SharedE-V2 uses a softmax-based routed component, whereas SharedE-V3 uses a sigmoid-based affinity score with normalization over the selected experts, allowing us to separate the effect of shared experts from the effect of the routing score.

**TC-MoE.**  TC-MoE augments the standard top-$K$ paradigm through ternary expert choice (Yan et al., 2025). Instead of treating an expert as only selected or not selected, it constructs an expanded routing space in which experts are associated with ternary choices from $-1, 0, 1$. This gives the router a richer set of activation patterns than ordinary top-$K$ selection. The method also uses auxiliary objectives tailored to the expanded routing space, including load-balancing and reward-style terms. In our benchmark, TC-MoE represents a structurally expressive routing design aimed at reducing redundant expert usage.

**MoE++.**  MoE++ augments the standard FFN expert pool with zero-computation experts (Jin et al., 2024). Instead of routing every token to a fixed number of FFN experts, it introduces zero, copy, and

constant experts, which respectively discard the input, copy the input as a shortcut, or replace it with a lightweight trainable-vector pathway. This allows different tokens to use different amounts and types of computation, while gating residuals help the router incorporate pathway information from previous layers. In our benchmark, MoE++ represents a heterogeneous routing design that targets a better trade-off between computational efficiency and model expressivity.

**Design dimensions.** Taken together, the seven algorithms vary along three principal axes: (i) the *scoring function* used by the router (softmax, sigmoid, or cosine-based normalization), (ii) the *selection policy* that determines which experts are activated, and (iii) the presence of *structural augmentation*, such as shared experts or zero-computation experts. Covering all three axes is important for LibMoE because it allows us to compare not only end-task performance, but also how different design choices affect routing stability, specialization, and expert utilization.

## B  Sensitivity to the Number of Active Experts

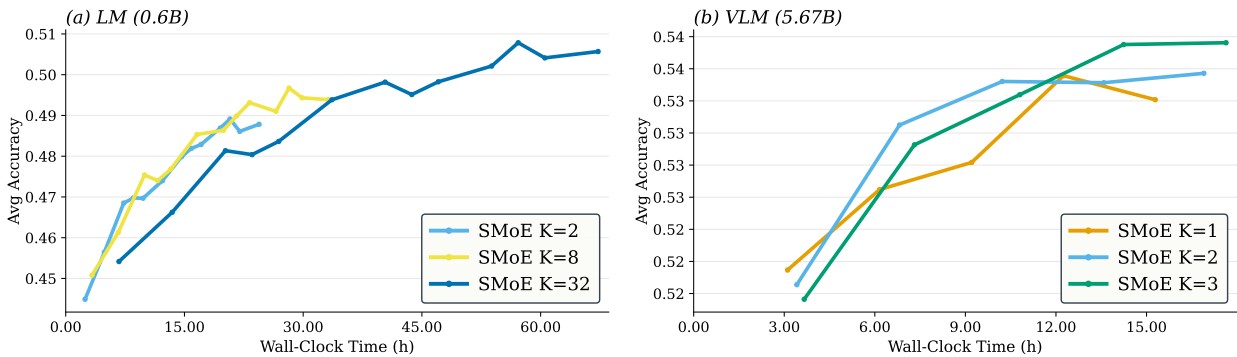

Figure 10: Average validation performance as a function of wall-clock time under different numbers of active experts $K$ for a standard SMoE model in language modeling (left) and vision-language training (right, 5.67B). For the VLM panel, training is performed on LLaVA-665K and validation performance is averaged over nine benchmarks, excluding MathVista and Hallusion Bench.

In this section, we provide a supplementary sensitivity analysis on the number of active experts. Our main benchmark fixes $K$ in each setting to keep comparisons across routing algorithms computationally controlled, so Figure 10 is intended primarily to illustrate how the training dynamics change when $K$ is varied for a representative SMoE model in language modeling and vision-language training.

In the language modeling setting, we compare $K \in \{2, 8, 32\}$. Within this setup, the curves suggest a trade-off between time-to-quality and the best validation performance reached later in training. Smaller or moderate routing budgets ($K=2$ and $K=8$) improve more quickly during the earlier stages, whereas the $K=32$ run progresses more slowly but continues improving over a longer horizon. Among the smaller settings, $K=8$ appears to offer a reasonable compromise in this experiment, while the largest setting finishes highest by the end of training.

In the vision-language setting, we compare $K \in \{1, 2, 3\}$. The dependence on $K$ appears weaker here than in the language-modeling case. Moving from $K=1$ to $K=2$ improves the trajectory in our run, while increasing from $K=2$ to $K=3$ yields a smaller difference. The $K=3$ curve ends slightly above $K=2$, but the two remain close for most of training.

Overall, these plots suggest that the effect of top-$K$ is regime-dependent in our benchmark. In the language-modeling setting, smaller $K$ values appear to improve training efficiency, whereas larger $K$ may help the final validation score when longer training is allowed. In the VLM setting, the gains beyond a small active-expert budget appear limited. We view these observations as descriptive rather than prescriptive, since the preferred choice of $K$ is likely to depend on factors such as scale, data, training budget, and deployment constraints.

## C Multi-Seed Results for the Main Benchmark Tables

To complement the single-run results in the main benchmark tables, we report multi-seed mean±std results for both the LLaVA-665K VLM setting and the small-model language-modeling setting. We use three random seeds: 42, 128, and 456. This presentation provides a more complete view of performance by showing both the central tendency and the run-to-run variation of each method. Overall, the multi-seed tables are broadly consistent with the main-text picture: under matched budgets, performance differences are generally modest, and the apparent ordering depends on the benchmark, the aggregate metric, and the training regime. We therefore treat this section as descriptive evidence rather than as support for a universally superior SMoE variant.

**Vision-language modeling (LLaVA-665K).** Table 3 provides the corresponding multi-seed results for the 665K sparse-upcycling benchmark. The overall spread is even smaller in this setting. XMoE attains the best mean on the two aggregate metrics (AVG Acc and AVG Rank), but only by a narrow margin, while the best mean on individual benchmarks is distributed across multiple methods. We therefore interpret the VLM multi-seed results conservatively: in this regime, the relative ordering of methods is benchmark-dependent, and no single variant separates cleanly from the others.

Table 3: Multi-seed mean±std results for the main VLM benchmark in the LLaVA-665K setting. Bold values indicate the best mean in each row; ties are boldfaced for all methods attaining the same best mean.

| Metric | SMoE | XMoE | $\sigma$-MoE | SharedE-V2 | SharedE-V3 | TC-MoE | MoE++ |
|---|---|---|---|---|---|---|---|
| AI2D ↑ | $64.78 \pm 0.73$ | $65.74 \pm 0.38$ | $65.71 \pm 0.61$ | $65.53 \pm 0.66$ | $65.71 \pm 0.20$ | $65.22 \pm 0.45$ | $\mathbf{65.83 \pm 0.69}$ |
| Text VQA ↑ | $41.57 \pm 0.26$ | $41.81 \pm 0.20$ | $41.78 \pm 0.29$ | $\mathbf{41.93 \pm 0.09}$ | $41.78 \pm 0.29$ | $40.82 \pm 0.41$ | $40.99 \pm 0.44$ |
| GQA ↑ | $61.36 \pm 0.28$ | $61.44 \pm 0.37$ | $61.70 \pm 0.07$ | $61.55 \pm 0.28$ | $\mathbf{61.85 \pm 0.51}$ | $61.34 \pm 0.17$ | $60.93 \pm 0.41$ |
| MM Bench ↑ | $\mathbf{72.45 \pm 0.42}$ | $72.11 \pm 0.35$ | $71.16 \pm 0.43$ | $71.62 \pm 0.61$ | $71.99 \pm 0.54$ | $71.25 \pm 0.30$ | $71.39 \pm 0.31$ |
| Hallusion Bench ↑ | $41.85 \pm 0.28$ | $42.27 \pm 0.42$ | $41.71 \pm 1.15$ | $42.13 \pm 0.58$ | $41.89 \pm 0.43$ | $\mathbf{42.41 \pm 0.30}$ | $41.99 \pm 0.95$ |
| Math Vista ↑ | $30.03 \pm 0.50$ | $30.73 \pm 0.78$ | $30.00 \pm 0.79$ | $30.77 \pm 0.91$ | $30.40 \pm 0.20$ | $30.20 \pm 0.90$ | $\mathbf{30.97 \pm 0.76}$ |
| MMMU ↑ | $42.33 \pm 0.59$ | $41.45 \pm 0.30$ | $41.48 \pm 0.36$ | $\mathbf{42.96 \pm 0.32}$ | $42.07 \pm 0.46$ | $41.85 \pm 1.00$ | $41.78 \pm 1.13$ |
| MMStar ↑ | $\mathbf{42.53 \pm 0.30}$ | $40.82 \pm 0.76$ | $41.40 \pm 1.36$ | $40.92 \pm 0.74$ | $41.71 \pm 0.04$ | $40.93 \pm 0.72$ | $40.79 \pm 0.60$ |
| Pope ↑ | $\mathbf{86.88 \pm 0.22}$ | $86.73 \pm 0.09$ | $86.66 \pm 0.32$ | $86.57 \pm 0.22$ | $\mathbf{86.88 \pm 0.41}$ | $86.66 \pm 0.14$ | $86.52 \pm 0.09$ |
| MME ↑ | $60.57 \pm 1.49$ | $\mathbf{60.86 \pm 0.55}$ | $60.71 \pm 0.70$ | $60.59 \pm 1.03$ | $60.77 \pm 0.33$ | $60.78 \pm 1.06$ | $59.63 \pm 0.51$ |
| MME RW ↑ | $32.41 \pm 0.33$ | $\mathbf{33.16 \pm 0.54}$ | $32.92 \pm 0.53$ | $32.31 \pm 0.60$ | $31.86 \pm 0.39$ | $32.27 \pm 0.99$ | $31.94 \pm 0.60$ |
| OCR Bench ↑ | $31.93 \pm 0.21$ | $\mathbf{33.00 \pm 0.35}$ | $32.83 \pm 0.31$ | $32.00 \pm 0.72$ | $32.40 \pm 0.35$ | $32.60 \pm 0.46$ | $31.43 \pm 0.40$ |
| AVG Acc ↑ | $50.72 \pm 0.17$ | $\mathbf{50.84 \pm 0.19}$ | $50.67 \pm 0.29$ | $50.74 \pm 0.04$ | $50.80 \pm 0.13$ | $50.53 \pm 0.35$ | $50.35 \pm 0.15$ |
| AVG Rank ↓ | $3.78 \pm 0.34$ | $\mathbf{3.19 \pm 0.10}$ | $3.74 \pm 0.67$ | $3.93 \pm 0.65$ | $3.58 \pm 0.78$ | $4.58 \pm 0.93$ | $5.20 \pm 0.16$ |

**Language modeling (0.15B).** Table 4 reports the multi-seed results for the main small-model language-modeling benchmark. Mean performance remains tightly clustered across most methods in this setting. TC-MoE achieves the best mean on both aggregate metrics (AVG Acc and AVG Rank), while MoE++, SharedE-V2, and $\sigma$-MoE remain close on the aggregate means. XMoE is weaker on the aggregate metrics in this particular setup, but the best mean on individual benchmarks is still distributed across several methods. We therefore view the language-modeling results as indicating a competitive leading group rather than a decisive separation between methods.

## D Analysis of a Hybrid Dense–Sparse Model

To examine whether a hybrid dense–sparse layout is beneficial in our setting, we conduct an additional experiment in the 5.67B-scale VLM benchmark on LLaVA-665K, comparing a hybrid architecture against its fully sparse counterpart. Following the design of DeepSeek-V3(Liu et al., 2024b), the hybrid variant keeps the first three FFN layers dense and replaces only the remaining FFN layers with MoE layers. As shown in Table 5, Hybrid SharedE-V3 underperforms the fully sparse SharedE-V3, despite providing only a modest reduction in training time (16h versus 16h47m). In our benchmark, this comparison suggests that the performance lost by removing MoE layers from the bottom of the network outweighs the corresponding efficiency gain.

Table 4: Multi-seed mean±std results for the main language-modeling benchmark in the small-model setting (0.15B). Bold values indicate the best mean in each row.

| Metric | SMoE | XMoE | $\sigma$-MoE | SharedE-V2 | SharedE-V3 | TC-MoE | MoE++ |
|---|---|---|---|---|---|---|---|
| PPL ↓ | $13.65 \pm 0.02$ | $14.28 \pm 0.26$ | $13.63 \pm 0.02$ | $13.58 \pm 0.08$ | $13.56 \pm 0.12$ | $13.47 \pm 0.04$ | $\mathbf{13.46 \pm 0.08}$ |
| LAMBADA ↑ | $\mathbf{25.70 \pm 0.39}$ | $23.97 \pm 0.85$ | $25.56 \pm 0.76$ | $25.25 \pm 0.98$ | $25.27 \pm 0.79$ | $25.67 \pm 0.31$ | $25.69 \pm 0.29$ |
| BLiMP ↑ | $\mathbf{77.31 \pm 1.15}$ | $75.74 \pm 0.75$ | $76.80 \pm 0.58$ | $77.13 \pm 0.25$ | $77.12 \pm 0.27$ | $76.96 \pm 0.39$ | $76.96 \pm 0.62$ |
| CBT ↑ | $84.45 \pm 0.24$ | $83.75 \pm 0.34$ | $84.39 \pm 0.43$ | $84.41 \pm 0.09$ | $84.39 \pm 0.26$ | $\mathbf{84.62 \pm 0.17}$ | $84.60 \pm 0.20$ |
| HellaSwag ↑ | $29.17 \pm 0.25$ | $29.02 \pm 0.29$ | $28.99 \pm 0.22$ | $29.14 \pm 0.21$ | $\mathbf{29.30 \pm 0.23}$ | $29.02 \pm 0.23$ | $29.15 \pm 0.13$ |
| PIQA ↑ | $58.14 \pm 0.19$ | $57.47 \pm 0.75$ | $58.90 \pm 0.52$ | $59.05 \pm 0.99$ | $58.71 \pm 0.38$ | $\mathbf{59.10 \pm 0.08}$ | $58.81 \pm 0.36$ |
| ARC-Easy ↑ | $32.61 \pm 0.12$ | $32.66 \pm 0.36$ | $32.97 \pm 0.28$ | $\mathbf{33.67 \pm 0.59}$ | $32.60 \pm 0.18$ | $33.16 \pm 0.52$ | $33.39 \pm 0.29$ |
| RACE ↑ | $30.26 \pm 0.15$ | $29.97 \pm 0.31$ | $\mathbf{30.98 \pm 0.06}$ | $30.95 \pm 0.28$ | $30.37 \pm 0.27$ | $30.62 \pm 0.15$ | $30.30 \pm 0.57$ |
| SIQA ↑ | $35.09 \pm 0.92$ | $35.93 \pm 0.49$ | $35.79 \pm 0.81$ | $35.36 \pm 0.20$ | $35.88 \pm 0.41$ | $\mathbf{36.17 \pm 0.43}$ | $35.43 \pm 0.51$ |
| CommonSenseQA ↑ | $24.62 \pm 0.13$ | $24.57 \pm 0.62$ | $24.87 \pm 0.05$ | $25.12 \pm 0.71$ | $24.98 \pm 0.49$ | $\mathbf{25.91 \pm 0.45}$ | $25.33 \pm 0.79$ |
| AVG Acc ↑ | $44.15 \pm 0.03$ | $43.67 \pm 0.18$ | $44.36 \pm 0.20$ | $44.45 \pm 0.18$ | $44.29 \pm 0.12$ | $\mathbf{44.58 \pm 0.06}$ | $44.41 \pm 0.18$ |
| AVG Rank ↓ | $4.70 \pm 0.15$ | $6.02 \pm 0.39$ | $3.83 \pm 0.50$ | $3.37 \pm 0.53$ | $3.75 \pm 0.48$ | $\mathbf{3.08 \pm 0.24}$ | $3.25 \pm 0.49$ |

Table 5: Comparison for the SHAREDE-V3 hybrid sparse-upcycling setting on LLaVA-665K. Hybrid SHAREDE-V3 denotes the dense precursor, and SHAREDE-V3 denotes the sparsely upcycled counterpart. Bold values mark the better score in each column.

| Model Variant | AI2D | Text VQA | GQA | MM Bench | Hallusion Bench | Math Vista | MMMU | MMStar | Pope | MME | MME RW | OCR Bench | AVG Acc ↑ |
|---|---|---|---|---|---|---|---|---|---|---|---|---|---|
| Hybrid SHAREDE-V3 | **66.22** | 41.44 | **61.73** | 71.22 | **42.17** | **30.80** | 40.78 | 40.91 | 86.78 | 59.08 | **32.00** | 32.10 | 50.43 |
| SHAREDE-V3 | 65.58 | **42.06** | 61.26 | **72.42** | 41.43 | 30.60 | **42.44** | **41.75** | **86.81** | **60.93** | 31.47 | **32.60** | **50.86** |

One plausible explanation is that early layers already play an important role in establishing token-dependent specialization. Dense FFNs apply the same MLP to every token, whereas MoE layers provide larger conditional capacity through token-wise expert selection. Replacing the lower MoE layers with dense ones may therefore weaken specialization from the start of the network and reduce the cumulative benefit of sparse routing in later layers.

Importantly, this observation is consistent with prior work (Zhong et al., 2024a; Lin et al., 2024a), which likewise reports that introducing experts only in the second half of the model underperforms applying experts across all layers. Taken together, these findings suggest that retaining dense early layers can reduce the benefits of sparse expert routing relative to a fully sparse allocation. Thus, while hybrid dense–MoE architectures remain an important design direction, our controlled experiment indicates that they should not be assumed to be universally superior, but instead evaluated under matched settings.

# E   Experiment Settings

## E.1   Vision Language Model

**Datasets.**   We adopt the vision-language pretraining task (Lu et al., 2019) and follow the CUMO framework (Li et al., 2024b) to upcycle the LLaVA model (Liu et al., 2023a), enabling systematic evaluation of various SMoE algorithms. In the initial **dense training stage**, we first initialize the MLP connector using the LLaVA-558K dataset (Liu et al., 2023a), then jointly train all three model components (image encoder, language model, and connector) on the ALLaVA dataset (Chen et al., 2024a). This yields a dense checkpoint that serves as the starting point for sparse upcycling.

In the subsequent SMoE training stage, we consider two training settings:

- LLaVA-665K (Liu et al., 2023a), a standard and widely used benchmark in the community;

- A hybrid dataset with 1M2 samples, constructed by combining LLaVA-665K with OneVision (Li et al., 2024a), where we uniformly sample 25% from each OneVision sub-benchmark to ensure broad domain coverage.

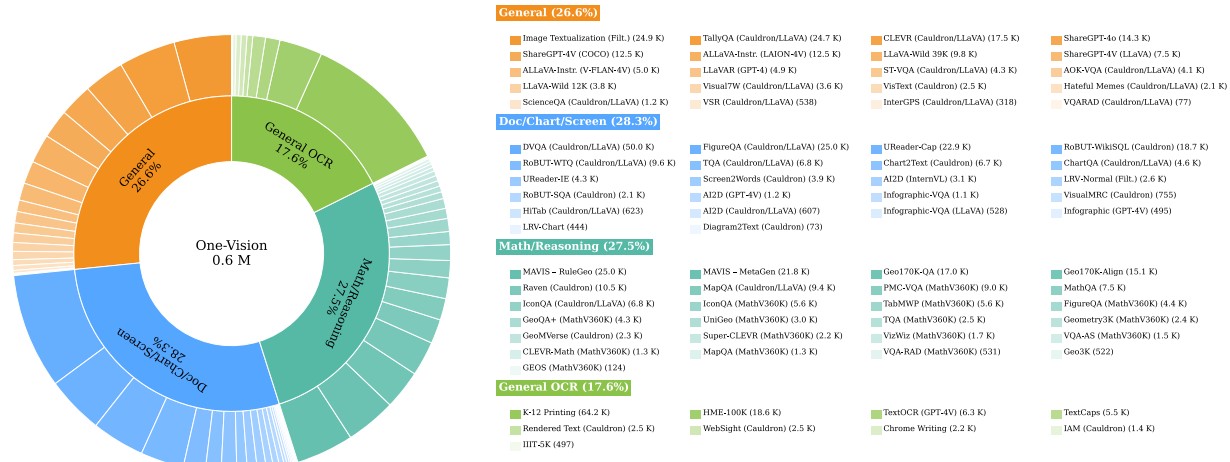

Figure 11: Visualization of dataset subsets from the OneVision dataset in LibMoE.

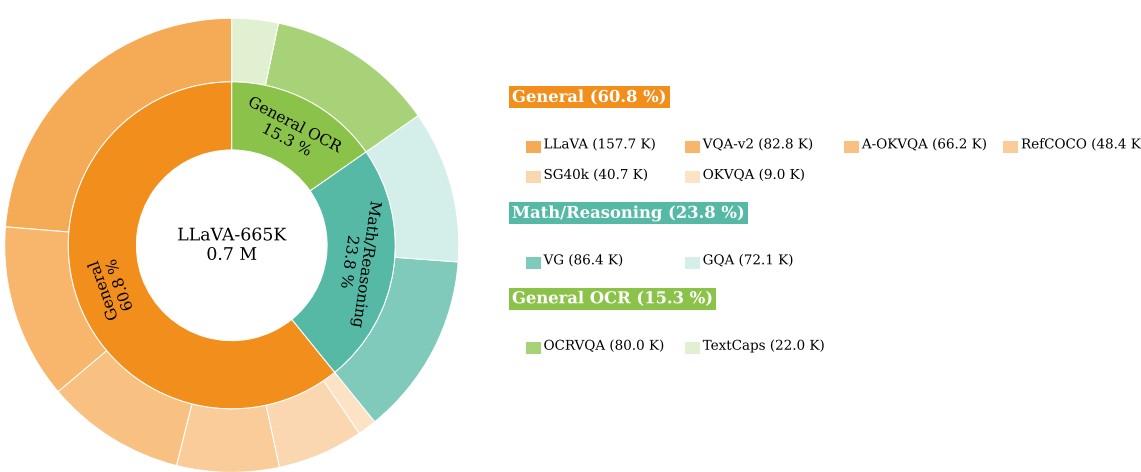

Figure 12: Visualization of dataset subsets from the LLAVA-665K dataset in LibMoE.

We visualize the data distribution across categories for both datasets in Figure 12 and Figure 11. All SMoE algorithms are trained using the same datasets and configurations to ensure fair benchmarking. For further details on dataset construction and training stage objectives, we refer readers to Liu et al. (2023a); Li et al. (2024b).

**Evaluation Benchmarks and Metrics**  LibMoE employs a set of popular benchmarks for vision-language models, including AI2D (Kembhavi et al., 2016), Text VQA Validation (Singh et al., 2019), GQA (Hudson & Manning, 2019), Hallusion Benchmark (Guan et al., 2023), MathVista (Lu et al., 2023), MMBenchEN (Liu et al., 2023b), MME (Fu et al., 2023), MMMU Validation(Yue et al., 2023), MMStar (Chen et al., 2024b), POPE (Li et al., 2023), MME-Real World(Zhang et al., 2024c) and OCR Bench(Liu et al., 2024c). We carefully choose these benchmarks to assess the model across several vision-language capabilities such as perception, reasoning, OCR, instruction following, and more. Beyond the performance on standard benchmarks, we analyze the algorithms holistically by evaluating their generalization throughout training, expert selection behaviors, and expert specialization, which we report in Section 5. For benchmarks requiring GPT-based evaluation, such as MathVista and HallusionBench, we use GPT-4o-mini, version 2024-07-18.

**Model Architecture.** We adopt PHI-3.5 MINI (Abdin et al., 2024) as the language model and SIGLIP-SO400M (Zhai et al., 2023) as the vision encoder two widely used backbones in recent multimodal systems. During the VIT stage, we upcycle the dense MLP blocks in both the vision encoder and connector into sparse MoE layers, each composed of $N_E=6$ experts with top-$K=3$ routing.

Additionally, Table 6 reports the parameter breakdown across key components. The full model contains approximately 5.67 billion parameters, including a 3.82B LLM backbone, a 1.75B vision encoder equipped with MoE layers, and 99M parameters dedicated to expert MLPs. To ensure fair comparison across routing methods, all models are configured under the same total parameter budget.

Table 6: Parameter counts for each major component. Total model size is 5.67B.

| Component | Parameters |
|---|---|
| Vision Encoder - MoE | 1.75B |
| MLP Connector - MoE | 0.099B |
| LLM | 3.82B |
| **Total** | 5.67B |

### E.2 Language Model Pretrain

**Dataset.** We conduct our experiments on the widely-used SlimPajama dataset (Soboleva et al., 2023), a high-quality corpus curated from RedPajama (Computer, 2023), specifically designed for training large language models (LLMs). SlimPajama has become a standard choice for open LLM research and has been employed in the development of several influential models, including TinyLlama (Zhang et al., 2024b) and BTLM (Dey et al., 2023), as well as in a range of recent empirical studies (Agarwalla et al., 2024; Gupta et al., 2023). Its adoption across diverse works underscores its value for benchmarking and advancing LLM research.

**Tokenizer and Model Architecture.** We adopt the SentencePiece tokenizer (Kudo & Richardson, 2018) with byte-pair encoding (BPE), which provides a balance between subword granularity and vocabulary efficiency. We generally follow Switch Transformer (Fedus et al., 2022) for our model architecture, adopting a standard Transformer backbone with sparsely activated Mixture-of-Experts (MoE) layers. Each MoE layer replaces the conventional feedforward sublayer and comprises $N$ expert networks (FFNs) and a router mechanism. Table 9 summarizes the comprehensive set of hyperparameters and configurations for both scales and different model variants evaluated in our experiments.

**Load Balancing Loss and Router Z-Loss.** We adopt the standard load balancing loss from Switch Transformer (Fedus et al., 2022) to penalize imbalances in routing decisions, thereby encouraging more uniform expert utilization. This auxiliary term has been shown to mitigate expert under-utilization and promote balanced load distribution, which improves both convergence and generalization in MoE models. In contrast, we do not incorporate the router z-loss in our experiments (Zoph et al., 2022). Following common practice, we set the weight of the load balancing loss to $\alpha = 0.01$. Additional experimental details and design considerations for the auxiliary load balancing loss are provided in Appendix H.4.

**Evaluation pipeline.** We evaluate our implemented models with the Perplexity score (PPL) and zero-shot performance with nine different downstream tasks: LAMBADA (Paperno et al., 2016), BLiMP (Warstadt et al., 2020), Children's Book Test (Hill et al., 2015), HellaSwag (Zellers et al., 2019), PIQA (Bisk et al., 2020), ARC-Easy (Clark et al., 2018), RACE (Lai et al., 2017), SIQA (Sap et al., 2019) and CommonSenseQA (Talmor et al., 2018). For LAMBADA, we use the detokenized version from OpenAI, and we evaluate the top-1 accuracy of the last word (it can span multiple tokens; here we use greedy decoding). For CBT, BLiMP, and RACE, we measure the accuracy of each task and report the average accuracy of the tasks.

# F Hyperparameter Setting

## F.1 Vision-Language Model

Table 7: Hyperparameter settings across the three training stages of PHI-3.5 MINI. MoE routing is only applied during the final VIT phase.

| Hyperparameter | PT | PFT | VIT |
|---|---|---|---|
| Learning rate | 1e-3 | 2e-6 | 4e-6 |
| Learning rate schedule | Cosine | Cosine | Cosine |
| Batch size per GPU | 64 | 6 | 4 |
| GPUs | 4×H100 | 4×H100 | 4×H100 |
| ZeRO optimization | ZeRO-2 | ZeRO-2 | ZeRO-3 |
| Optimizer | AdamW | AdamW | AdamW |
| MLP parameters | Trained | Trained | Trained |
| Vision encoder | Frozen | Trained | Trained |
| Language model | Frozen | Trained | Trained |
| MoE blocks | No | No | Yes |
| Balance loss coefficient | 0.0 | 0.0 | 0.01 |
| Z-loss coefficient | 0.0 | 0.0 | 0.001 |
| Maximum tokens | 2048 | 2048 | 2048 |

**Hyperparameter Settings and Training Stages.** Table 7 summarizes the key hyperparameter configurations across the three sequential training stages of PHI-3.5 MINI: Pretraining (PT), Pre-FineTuning (PFT), and Visual Instruction Tuning (VIT). All stages are trained on 4×H100 GPUs with consistent token lengths, optimizers, and learning rate schedules to ensure comparability. The Mixture-of-Experts (MoE) routing mechanism is activated only during the VIT stage, where multimodal specialization is required. Following the initialization strategy used in the official GPT-2 implementation,[1] router parameters are sampled from a normal distribution $\mathcal{N}(0, 0.02^2)$, i.e., with standard deviation 0.02, using a fixed random seed of 42 to ensure reproducibility. All expert parameters are trained during VIT, while only the router is initialized from scratch. This design yields a controlled and fair evaluation protocol across different sparse routing strategies.

Table 8: MoE configuration details across different methods, where $N$ denotes the total number of experts, $K$ the number of routed (active) experts per token, and $N_s$ the number of shared experts.

| MoE Method | $N$ | $K$ | $N_s$ |
|---|---|---|---|
| SMoE | 6 | 3 | 0 |
| SharedE | 6 | 2 | 1 |
| MoE++ | 8 | 3 | 0 |
| TC-MoE | 15 | 3 | 0 |

---

[1] https://github.com/openai/gpt-2

**MoE Architecture Configurations.** Table 8 summarizes the MoE configurations used across different methods, where $N$ denotes the total number of experts, $K$ the number of routed (active) experts per token, and $N_s$ the number of shared experts. For standard routing-based methods, including SMoE, $\sigma$-MoE, and XMoE, we adopt a common configuration with $N = 6$ experts and top-$K = 3$ routing. For shared-expert architectures, we follow the VLM setting with one shared expert ($N_s = 1$) and reduce the number of routed experts to $K = 2$ to maintain comparable computational cost. For MoE++, we follow Jin et al. (2024) and augment the expert set by introducing zero-computation experts, resulting in a total of $N = 8$ experts while keeping $K = 3$. For TC-MoE, we follow Yan et al. (2025), where the effective expert pool is expanded via ternary compositions, yielding a total of $2N + K$ experts; in our configuration, this corresponds to $N = 15$ with $K = 3$.

## F.2 Language Modeling

Table 9: Comprehensive Model Configurations for Pre-train LLM. SMoE refers to settings applied for Vanilla SMoE, $\sigma$-MoE and XMoE, whereas SharedE corresponds to configurations used for SharedE-V2 and SharedE-V3 models.

| Scale | Model | # params | # act. params | # trained tokens | $d_{\mathrm{model}}$ | H | $d_{\mathrm{head}}$ | $N$ | $K$ | $N_s$ | Expert dim |
|---|---|---|---|---|---|---|---|---|---|---|---|
| Small | SMoE | 0.15B | 36M | 6.55B | 512 | 12 | 82 | 66 | 8 | 0 | 128 |
| | SharedE | | | | | | | 66 | 6 | 2 | |
| | MoE++ | | | | | | | 66 + 8 | 8 | 0 | |
| | TC-MoE | | | | | | | 66 * 2 + 8 | 8 | 0 | |
| Large | SMoE | 0.68B | 131M | 26.2B | 1024 | 16 | 128 | 66 | 8 | 0 | 256 |
| | SharedE | | | | | | | 66 | 6 | 2 | |
| | MoE++ | | | | | | | 66 + 8 | 8 | 0 | |
| | TC-MoE | | | | | | | 66 * 2 + 8 | 8 | 0 | |

**Hyperparameter Settings and Training Stages.** Table 9 summarizes the key hyperparameters, covering model dimensionality, number of attention heads, expert counts, and routing strategies. The small-scale setting processes 6.55B tokens with model dim $d_{model} = 512$ and number of attention heads $H = 12$, while the large-scale setting extends to 26.2B tokens with $d_{model} = 1024$ and $H = 16$. Each variant differs in expert number ($N$), routing capacity ($K$), and warmup strategies, with expert dimensions set to 128 for small and 256 for large models. These standardized yet diverse configurations ensure a balanced comparison across MoE algorithms while reflecting realistic large-scale training regimes.

Table 10 details the training hyperparameters, where both scales are optimized with AdamW using a cosine learning rate schedule, gradient clipping, and mixed-precision training on 4×H100 GPUs. While the small-scale models are trained for 100k steps on 6.55B tokens, large-scale models extend to 400k steps on 26.2B tokens, with warm-up and stronger gradient clipping applied to ensure stability. Together, these settings provide a consistent yet scalable foundation for benchmarking diverse MoE algorithms.

**MoE Architecture Configurations.** Table 9 also specifies the MoE architecture configurations used in the language modeling pre-training setting, where $N$ denotes the total number of experts, $K$ the number of routed (active) experts per token, and $N_s$ the number of shared experts. For standard routing-based methods (SMoE, $\sigma$-MoE, and XMoE), we adopt a common configuration with $N = 66$ experts and top-$K = 8$ routing. For shared-expert variants (SharedE-V2/V3), we set $N_s = 2$ shared experts and reduce routed capacity to $K = 6$ to maintain comparable compute. For MoE++, following Jin et al. (2024), we augment the expert pool by adding zero-computation experts, resulting in $N = 66 + 8$ while keeping $K = 8$. For TC-MoE, following Yan et al. (2025), the effective expert pool is expanded via ternary compositions, yielding $2N + K$ experts; under our configuration, this corresponds to $N = 66 \times 2 + 8$ with $K = 8$. Across both model scales,

Table 10: Training hyperparameter settings for LibMoE across two model scales (0.15B and 0.68B parameters) in the language modeling task.

| Hyperparameter | 0.15B | 0.68B |
|---|---|---|
| Learning rate | 2.5e-4 | 2.5e-4 |
| LR schedule | Cosine | Cosine |
| $N_{warmup}$ | 0 | 4000 |
| Min LR multiplier | 0.1 | 0.1 |
| Optimizer | AdamW | AdamW |
| Weight decay | 0.01 | 0.01 |
| Gradient clip ($\kappa$) | 0.1 | 0.25 |
| Dropout | No | No |
| Batch size (per device) | 16 | 16 |
| Total batch size | 64 | 64 |
| Sequence length | 1024 | 1024 |
| Training steps | 100k | 400k |
| Validation ratio | 0.5% | 0.5% |
| Precision | AMP (fp16) | AMP (fp16) |
| GPUs | $4 \times$ H100 | $4 \times$ H100 |

we keep the expert dimension fixed (128 for 0.15B and 256 for 0.68B) to ensure a controlled comparison across MoE designs.

## G    Comparison Between Dense and SMoE Models

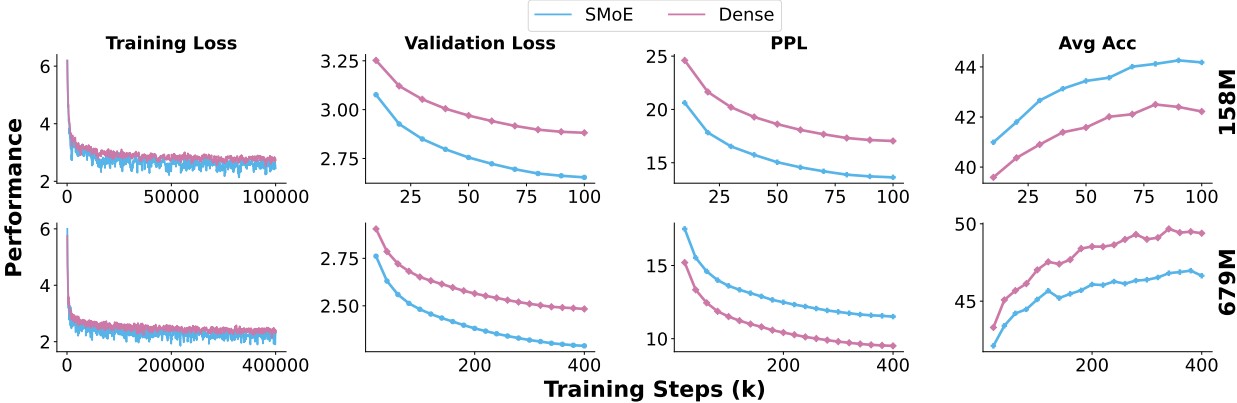

Figure 13: Training benchmark curves comparing Dense and SMoE models during language model pre-training (0.67B parameters).

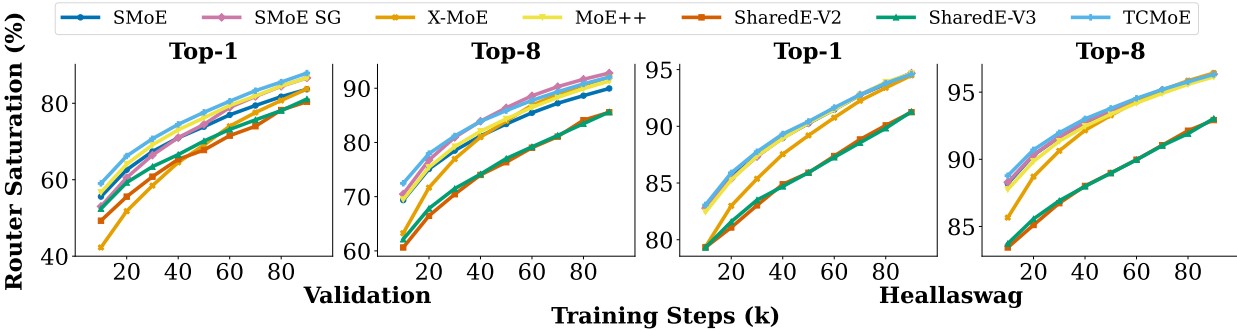

Figure 14: Router Saturation across methods during training on the language modeling task. We present both Top-1 and Top-8 routing results for small (0.15B) and large (0.68B) models, illustrating the progression of router convergence across model scales and expert selection strategies.

Table 11: Performance comparison between dense and SMoE models for language model pre-training, evaluated on a small-scale model (0.15B parameters) and a large-scale model (0.68B parameters). PPL denotes perplexity; lower values indicate better performance.

| | MoE Method | PPL ↓ | LAMBADA | BLiMP | CBT | Hella Swag | PIQA | ARC-Easy | RACE | SIQA | Common SenseQA | AVG ↑ |
|---|---|---|---|---|---|---|---|---|---|---|---|---|
| Small Model (0.15B) | Dense (36M) | 17.04 | 19.74 | 73.48 | 81.03 | 27.56 | 55.82 | 32.09 | 29.36 | 35.72 | 25.14 | 42.22 |
| | SMoE | 13.63 | 25.27 | 77.71 | 84.18 | 29.43 | 57.94 | 32.68 | 30.11 | 35.62 | 24.65 | 44.18 |
| Large Model (0.68B) | Dense (131M) | 11.51 | 31.00 | 77.43 | 87.45 | 31.90 | 61.37 | 35.14 | 31.58 | 36.90 | 27.03 | 46.64 |
| | SMoE | 9.51 | 37.13 | 80.47 | 89.83 | 37.49 | 64.36 | 38.22 | 33.03 | 37.41 | 26.54 | 49.39 |

Figure 13 and Table 11 presents a comparative analysis between dense baselines and SMoE models in the language model pre-training setting. Across all reported metrics, SMoE models consistently achieve stronger performance during the early stages of training, indicating faster optimization and more efficient utilization of model capacity compared to dense counterparts.

# H   Additional Analysis

## H.1   Router Saturation

**Router Saturation** - first introduced in OLMoE (Muennighoff et al., 2024) - quantifies the proportion of overlapping activated experts between an intermediate checkpoint at training step $t$ and the final checkpoint. This metric serves as an indicator of the router's convergence dynamics throughout training. Higher router saturation values indicate greater alignment in expert selection, signifying that the router's decisions are becoming increasingly consistent with its final checkpoint. Consequently, router saturation provides insight into the convergence of expert assignments of routing strategies during training. The formal definition and formula are defined in the Appendix I.4.

As shown in Figure 14, router saturation for all evaluated methods rises sharply during training, with most surpassing 60%, and even reaching over 85% on the HellaSwag benchmark, within the first 10% of training under top-8 selection. In contrast, top-1 selection exhibits slightly slower convergence, indicating a more gradual stabilization of expert assignments. Notably, XMoE stands out as an outlier, converging more slowly than other variants—a trend consistent with its relatively lower performance reported in Table 2. Overall, this early stabilization behavior aligns with prior work (Muennighoff et al., 2024; Xue et al., 2024; Nguyen et al., 2025b; Kang et al., 2025), suggesting a general tendency for MoE routers to converge rapidly toward stable expert assignments. Such early convergence is also consistent with the learning strategy advocated by Stable MoE (Dai et al., 2022), which aims to mitigate fluctuations in expert allocation during training.

Table 12: Performance variation under different router temperatures across MoE algorithms. We evaluate the impact of router temperature ($\tau$) on expert cooperation and competition in both small (0.15B) and large (0.68B) models, using two language understanding tasks (BLiMP and HellaSwag). Under this parameterization, lower $\tau$ sharpens routing and increases competition, whereas higher $\tau$ smooths routing and promotes cooperation.

| | $\tau$ | Task | SMoE | SMoE SG | XMoE | SharedE-V2 | SharedE-V3 | TC-MoE | MoE++ | AVG $\Delta$ |
|---|---|---|---|---|---|---|---|---|---|---|
| Small Model (0.15B) | 1.0 | BLIMP | 77.71% | 76.75% | 76.53% | 77.37% | 77.20% | 76.91% | 77.23% | |
| | | HellaSwag | 29.43% | 29.15% | 29.34% | 29.38% | 29.38% | 29.27% | 29.28% | |
| | 10.0 | BLIMP | 69.64%↓8.07 | 68.52%↓8.24 | 76.23%↓0.30 | 56.87%↓20.5 | 55.41%↓21.8 | 63.73%↓13.2 | 67.98%↓9.25 | ↓11.6% |
| | | HellaSwag | 29.55%↑0.12 | 29.41%↑0.26 | 29.31%↓0.03 | 28.75%↓0.63 | 28.79%↓0.59 | 29.42%↑0.15 | 29.40%↑0.12 | ↓0.09% |
| | 0.1 | BLIMP | 63.89%↓13.8 | 51.85%↓24.9 | 76.47%↓0.06 | 70.30%↓7.07 | 69.65%↓7.56 | 63.57%↓13.3 | 64.14%↓13.1 | ↓11.4% |
| | | HellaSwag | 27.82%↓1.60 | 25.03%↓4.11 | 28.69%↓0.65 | 28.91%↓0.47 | 29.22%↓0.16 | 27.41%↓1.85 | 27.66%↓1.61 | ↓1.49% |
| Large Model (0.68B) | 1.0 | BLIMP | 80.47% | 81.08% | 80.38% | 80.98% | 81.28% | 81.21% | 80.88% | |
| | | HellaSwag | 37.49% | 37.52% | 37.19% | 37.14% | 37.32% | 37.95% | 37.70% | |
| | 10.0 | BLIMP | 73.96%↓6.51 | 76.56%↓4.52 | 80.52%↑0.13 | 75.05%↓5.93 | 76.45%↓4.83 | 66.34%↓14.9 | 73.37%↓7.51 | ↓6.29% |
| | | HellaSwag | 34.37%↓3.12 | 35.30%↓2.22 | 36.72%↓0.48 | 33.44%↓3.70 | 34.92%↓2.40 | 32.12%↓5.84 | 33.98%↓3.72 | ↓3.07% |
| | 0.1 | BLIMP | 67.94%↓12.5 | 71.24%↓9.84 | 79.13%↓1.25 | 71.69%↓9.29 | 74.78%↓6.50 | 66.33%↓14.9 | 68.03%↓12.9 | ↓9.59% |
| | | HellaSwag | 32.57%↓4.92 | 35.42%↓2.10 | 36.44%↓0.76 | 33.72%↓3.42 | 35.18%↓2.14 | 32.52%↓5.43 | 33.17%↓4.53 | ↓3.33% |

## H.2 Cooperation or Competition

To characterize expert behavior when contributing to the creation of outcomes, we analyze the performance change that occurs when we adjust the router's temperature. Under the parameterization below, a low router temperature makes the expert weights more peaky, indicating stronger competition between experts. In contrast, a high temperature makes expert weights more uniform, indicating more cooperative behavior. For sigmoid gating, the same trend holds: decreasing $\tau$ pushes gate activations closer to $\{0, 1\}$, whereas increasing $\tau$ moves them closer to 0.5. To be more specific, if the logit of expert $k$ is $s_k$, our router is adjusted as follows:

$$g(s_k) = \begin{cases} \frac{\exp(s_k/\tau)}{\sum_{j=1}^{N_E} \exp(s_j/\tau)}, & \textit{if softmax router} \\ \sigma(s_k/\tau), & \textit{if sigmoid router} \end{cases} \tag{2}$$

Table 12 summarizes the effects of varying the router's temperature parameter on model performance for both small (0.15B) and large (0.68B) model sizes across two representative language understanding tasks. We observe that deviating from the original temperature used during training generally results in performance degradation across both small and large model scales. In most settings, applying a low temperature ($\tau = 0.1$), which induces sharper and more competitive routing, leads to the larger performance drop, although a few cases—such as XMoE and MoE++ on the small model—remain comparatively robust. Increasing the temperature to $\tau = 10.0$, which encourages flatter and more cooperative routing, also often hurts performance, but is typically less damaging than over-sharpening the router. Interestingly, XMoE demonstrates greater robustness to temperature changes compared to other variants.

Overall, these findings suggest that most existing MoE architectures operate best near the training temperature and, when perturbed, are generally more resilient to enhanced cooperation (higher temperature) than to heightened competition (lower temperature). We leave a deeper investigation of the underlying mechanisms driving this cooperation-competition trade-off to future work.

## H.3 Expert Co-Activation Over Time

To further examine expert interaction dynamics, we analyze the evolution of expert co-activation (ECA) patterns throughout training. Following Muennighoff et al. (2024), ECA measures how frequently two experts are activated together, normalized by the total number of activations of one expert (see Appendix I.5 for the formal definition). Higher ECA values indicate persistent co-utilization between expert pairs.

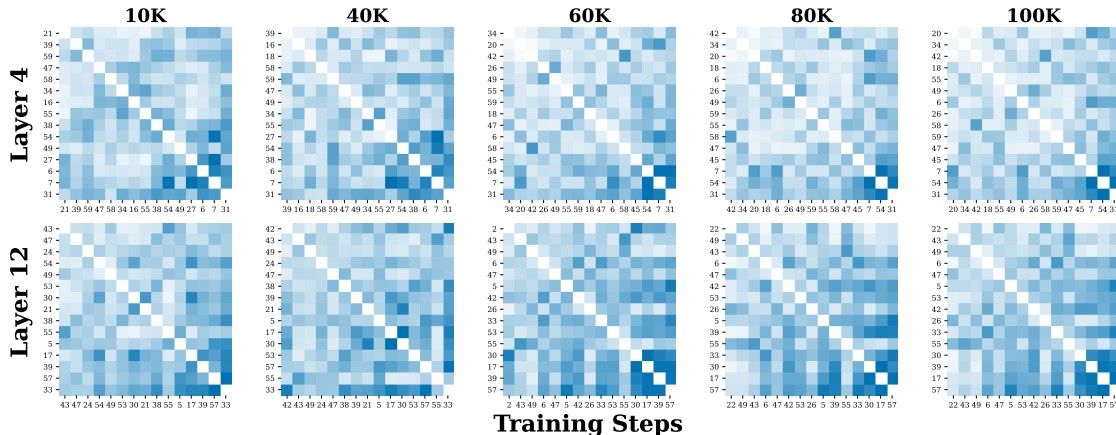

Figure 15: Expert co-activation (ECA) matrices for a Vanilla SMoE (0.15B) on the language modeling task, shown at Layers 4 and 12 across training checkpoints (10K–100K steps) on the validation set.

Figure 15 shows the ECA matrices for Layer 4 and Layer 12 of a small (0.15B) Vanilla SMoE model at multiple training checkpoints. Across training, the identity of the most strongly co-activated expert pairs remains largely unchanged, with only modest fluctuations in co-activation strength. This consistency suggests that expert collaboration structures emerge early in training and remain stable thereafter, indicating limited reorganization of expert interactions during later optimization.

We observe similar stability in expert co-activation patterns across other SMoE variants, including XMoE, SMoE-Sigmoid, DeepSeek-V2, DeepSeek-V3, TCMoE, and MoE++. Corresponding ECA visualizations are provided in Figure 20, Figure 21, Figure 22, Figure 25, Figure 24, and Figure 23. Collectively, these results indicate that, despite differences in routing mechanisms and architectural refinements, expert co-activation relationships in SMoE-style models are remarkably stable over training.

## H.4 Load Balancing Loss and Router Z-loss Experiment

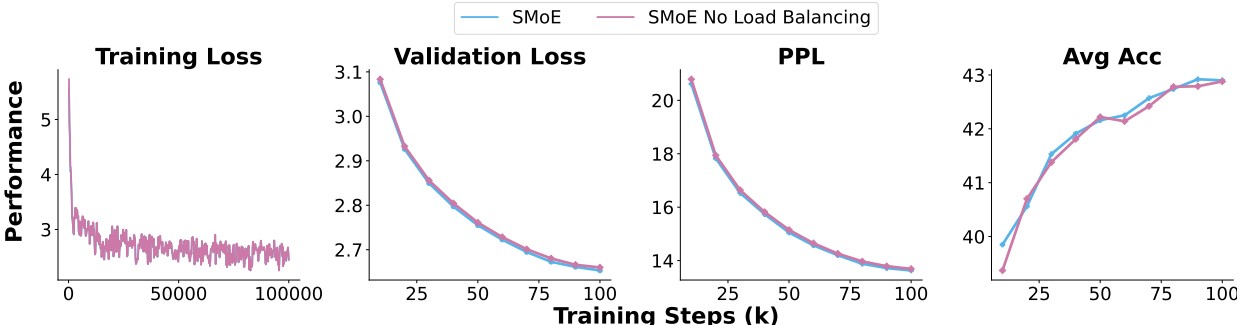

Figure 16: Benchmark curves during training in language modeling tasks for models with 0.15B parameters with and without load balancing loss.

Table 13: Performance comparisons in the impact of load balancing loss and router z-loss of small model (0.15B parameters) in language modeling task.

| MoE Method | PPL ↓ | LAM BADA | BLiMP | CBT | Hella Swag | PIQA | ARC-Challenge | RACE | SIQA | Common SenseQA | Average |
|---|---|---|---|---|---|---|---|---|---|---|---|
| SMoE (0.01 lb) | 13.63 | 25.27% | 77.71% | 84.18% | 29.43% | 57.94% | 21.20% | 30.11% | 35.62% | 24.65% | **42.90%** |
| SMoE (no-lb) | 13.69 | 24.90% | 76.81% | 84.13% | 29.38% | 57.51% | 21.63% | 30.26% | 35.67% | 25.63% | 42.88% |
| SMoE (0.01 lb + 0.001 z-loss) | 13.62 | 25.49% | 77.55% | 84.23% | 29.11% | 58.71% | 21.89% | 29.63% | 35.21% | 23.91% | 42.86% |

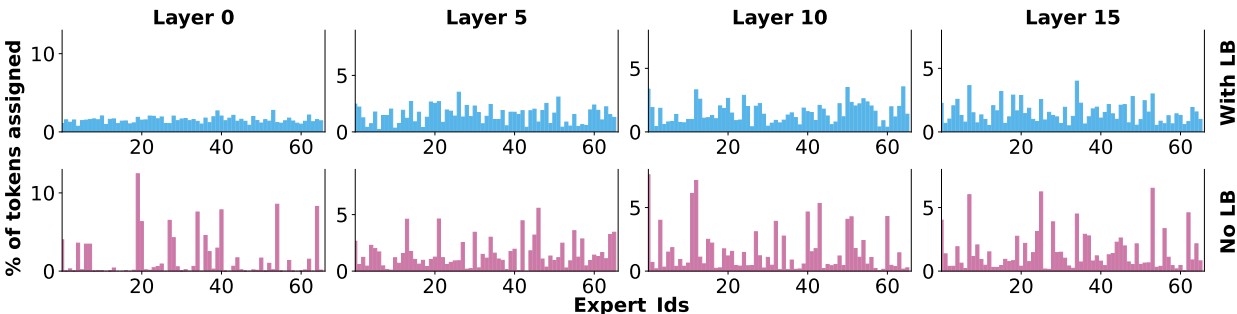

Figure 17: Expert selection ratio of SMoE model with and without load balancing loss in language modeling tasks (0.15B parameters model size).

Figure 16 and Figure 17 illustrate the effect of the load balancing loss on both training dynamics and expert assignment in Mixture-of-Experts models. The results demonstrate that incorporating load balancing loss contributes to more stable training and improved overall performance. Specifically, Figure 17 highlights how the load balancing loss promotes a more uniform distribution of tokens across experts, mitigating the risk of expert under-utilization.

Beyond load balancing, (Zoph et al., 2022) introduced the router z-loss as an additional regularization strategy to further stabilize MoE training. To assess the practical impact of these auxiliary losses, we report in Table 13 a comparison of model performance under different loss configurations for language-model pretraining. Our results indicate that applying the router z-loss does not yield a performance improvement in this setting; in fact, it slightly degrades accuracy relative to using only the load balancing loss. Therefore, we opt not to include the router z-loss in our final language-model experimental setup.

# I    Definitions and Formulations of Analysis Metrics

## I.1    Total Score Change under Routing Perturbations: DropTop1 vs. DropTop1&2

Let $\mathcal{D} = \{D_i\}_{i=1}^{|\mathcal{D}|}$ denote the set of evaluation benchmarks. For a given MoE model, let $M_i \in \mathbb{R}$ be its original evaluation score (e.g., accuracy; higher is better) on benchmark $D_i$. After applying a routing perturbation $\pi \in \{DropTop1, DropTop1\&2\}$ at inference time, we obtain the perturbed score $\hat{M}_i^\pi \in \mathbb{R}$.

We define the aggregate performance change (reported as $\Delta$ Total Score in Figure 4) as:

$$\Delta \text{TotalScore}^\pi = \sum_{i=1}^{|\mathcal{D}|} \left( \hat{M}_i^\pi - M_i \right). \tag{3}$$

Negative values indicate that the routing perturbation leads to a net performance degradation relative to the original routing configuration, whereas positive values indicate that the perturbed routing achieves higher aggregate performance than the original model. In the latter case, the improvement suggests that the original routing decision may have been suboptimal, and that alternative expert assignments can better exploit the model's representational capacity.

## I.2    Expert Entropy Allocation

We investigate the behaviors of the expert selection mechanism by exploring how often each expert is selected in the MME benchmark. To this end, we analyze the frequency of the expert selection across different subtasks to gain insights into the specialization behavior of each expert. Given an MoE algorithm with $N$ experts and $L$ layers, the selection frequency of each expert $i$ at a given layer $l$ is denoted as $\text{freq}_i^{(l)}$ ($i = 1, 2, \ldots, N$ and $l = 1, 2, \ldots, L$). Note that this selection frequency is counted across all samples in the benchmark, in this case we choose to be MME. Then, the entropy $H^{(l)}$ at each layer $l$ is calculated by integrating the probability of selecting expert $i$ into Shannon's entropy formula as follows:

$$EAE^{(l)} = \frac{-\sum_{i=1}^{N} \left( \frac{\text{freq}_i^{(l)}}{\sum_{j=1}^{N} \text{freq}_j^{(l)}} \right) \log_2 \left( \frac{\text{freq}_i^{(l)}}{\sum_{j=1}^{N} \text{freq}_j^{(l)}} \right)}{\log_2(N)},$$ (4)

where:

- $\text{freq}_i^{(l)}$: The number of times expert $i$ is selected at layer $l$.

- $N$: The total number of experts in the MoE algorithms.

- $\sum_{j=1}^{N} \text{freq}_j^{(l)}$: The total number of expert selections at layer $l$.

- $EAE^{(l)}$: The entropy value at layer $l$, measuring the uncertainty or diversity in expert selections.

With $EAE$, we can measure the frequency of expert selections across all layers $H_{EAE}$. Importantly, high $EAE$ Indicates a balanced expert utilization across all layers, where the model tends to distribute selections evenly among all experts. In contrast, low $EAE$ suggests a concentrated usage of a few experts in most layers, indicating specialization or a preference for certain experts.

### I.3    Router Margin

Router Margin is a metric that quantifies the dominance of the highest-scoring expert in a Mixture-of-Experts (MoE) routing decision. It is defined as the difference between the top-1 and top-2 gating scores:

$$\text{Router Margin}(l) = \frac{1}{N} \sum_{i=1}^{N} \left( top1(x_i) - top2(x_i) \right),$$

Where:

- RouterMargin($l$): Router margin at layer $l$

- $N$: The total number of tokens in the dataset.

- $x_i$: $i$-th input token

- $top1(x_i)$: top-1 routing score of input $x_i$

- $top2(x_i)$: top-2 routing score of input $x_i$

Router Margin provides a quantitative measure of how decisively the router selects the top expert relative to alternatives. A larger margin indicates that the router strongly favors the top-1 expert, reflecting more confident and specialized routing, whereas a smaller margin implies greater ambiguity and potential overlap among experts. This metric is thus a valuable diagnostic tool for understanding the dynamics of expert dominance and the evolution of routing confidence during training.

### I.4    Router Saturation

In formal terms, router saturation is the proportion of expert activations at some intermediary checkpoint at time $t$ that matches the expert IDs activated at some final checkpoint $T$ over the same dataset:

$$\text{RouterSaturation}(t) = \frac{1}{N} \sum_{i=1}^{N} \frac{\left| \mathcal{E}_i^{(t)} \cap \mathcal{E}_i^{(T)} \right|}{k},$$ (5)

Where:

- $N$: The total number of tokens in the dataset.

- $k$: The number of top-k experts activated per input token.

- $\mathcal{E}_i^{(t)}$: The set of $k$ experts activated for the $i$-th token at the $t$-th checkpoint.

- $\mathcal{E}_i^{(T)}$: The set of $k$ experts activated for the $i$-th token at the final checkpoint $T$.

- $\left|\mathcal{E}_i^{(t)} \cap \mathcal{E}_i^{(T)}\right|$: The number of common experts activated for the $i$-th token between the $t$-th and final checkpoints $T$.

Router saturation provides a quantitative measure of how early the routing decisions converge during training. A saturation value of 100% indicates that the router at an intermediate checkpoint routes to the same set of experts as at the final checkpoint. High saturation values at early checkpoints reflect early convergence in expert selection, indicating that the router has rapidly settled into a stable assignment pattern. In contrast, low saturation values suggest ongoing exploration or adaptation in expert allocations, signaling that the routing mechanism is still undergoing significant adjustments.

## I.5 Experts Co-activation

We define expert co-activation as the proportion of times two specific experts, $E_i$ and $E_j$, are simultaneously activated out of the total number of activations of one of those experts:

$$\text{Expert co-activation}(E_i, E_j) = \frac{N_{E_i, E_j}}{N_{E_i}}, \tag{6}$$

where:

- $E_i$: The first expert.

- $E_j$: The second expert.

- $N_{E_i, E_j}$: The number of times experts $E_i$ and $E_j$ are activated together.

- $N_{E_i}$: The total number of times expert $E_i$ is activated.

A co-activation of 100% indicates that if $E_i$ is activated, $E_j$ is also always activated. A value of 0% indicates that the experts never co-occur. If multiple expert pairs have high co-activation, it may suggest that these experts could be merged, benefiting less from keeping them separate. In a distributed setup, we could place highly co-activated experts on the same device to reduce communication costs during model inference.

## J   Training Time and Resource Allocation

### J.1   Training Time and Resource Usage

Table 14 reports the training time and GPU resource allocation across all experimental configurations. In the VLM setting, SHAREDE-V2 and SHAREDE-V3 are consistently the most training-efficient methods, completing OneVision training roughly two hours faster than the competing approaches. This advantage stems from their shared-expert design, which reduces per-token routing computation. The router therefore scores only the non-shared experts, i.e., $N$ minus the number of shared experts, lowering routing overhead and improving end-to-end efficiency. Importantly, this runtime reduction does not come at the cost of model quality. As shown in Table 1, both variants remain highly competitive and, in several cases, achieve the best overall performance on OneVision. For language-model pretraining, however, we use the CVMM Triton kernel Csordás et al. (2024), which substantially accelerates sparse MoE computation. As a result, runtime differences in this setting are influenced more by kernel-level optimization than by architectural choices such as SHAREDE-V2 or SHAREDE-V3. We therefore view training-time comparisons as most informative in the VLM setting, where the observed differences more directly reflect methodological design choices.

Table 14: Training Time and GPU Resource Allocation across all Experimental Settings.

| | Model | | Training Time (hours) | Resource Allocation |
|---|---|---|---|---|
| VLM | Pre-Training | | 2h35m | 4xH100 |
| | Pre-FineTuning | | 16h | 4xH100 |
| | Visual Instruction Tuning LLAVA-665K | SMoE | 17h19m | 4xH100 |
| | | XMoE | 17h59m | 4xH100 |
| | | $\sigma$-MoE | 17h32m | 4xH100 |
| | | SharedE-V2 | 16h29m | 4xH100 |
| | | SharedE-V3 | 16h47m | 4xH100 |
| | | TC-MoE | 18h01m | 4xH100 |
| | | MoE++ | 18h50m | 4xH100 |
| | Visual Instruction Tuning OneVision / 1M2 samples | SMoE | 33h01m | 4xH100 |
| | | XMoE | 34h27m | 4xH100 |
| | | $\sigma$-MoE | 33h33m | 4xH100 |
| | | SharedE-V2 | 31h19m | 4xH100 |
| | | SharedE-V3 | 31h39m | 4xH100 |
| | | TC-MoE | 33h47m | 4xH100 |
| | | MoE++ | 35h40m | 4xH100 |
| Language Modeling | 0.15B parametes | Dense | 5h47m | 4xH100 |
| | | SMoE | 6h15m | 4xH100 |
| | | XMoE | 6h21m | 4xH100 |
| | | $\sigma$-MoE | 6h12m | 4xH100 |
| | | SharedE-V2 | 6h25m | 4xH100 |
| | | SharedE-V3 | 6h23m | 4xH100 |
| | | TC-MoE | 6h10m | 4xH100 |
| | | MoE++ | 6h17m | 4xH100 |
| | 0.68B parametes | Dense | 41h30m | 4xH100 |
| | | SMoE | 42h11m | 4xH100 |
| | | XMoE | 42h54m | 4xH100 |
| | | $\sigma$-MoE | 41h57m | 4xH100 |
| | | SharedE-V2 | 43h01m | 4xH100 |
| | | SharedE-V3 | 43h09m | 4xH100 |
| | | TC-MoE | 41h38m | 4xH100 |
| | | MoE++ | 42h22m | 4xH100 |

### J.2 Peak GPU Memory Usage and Per-Sample Inference Latency

Table 15 reports the peak GPU memory usage observed during training for each SMoE method, together with the average per-sample inference latency, in the 5.67B VLM setting. These measurements provide a more complete practical comparison by showing how each routing design affects both memory footprint and inference efficiency under matched settings.

Table 15: Peak GPU memory usage during training and per-sample inference latency across SMoE methods in the 5.67B VLM setting.

| Method | Peak GPU Memory Usage | Inference Latency (s/sample) |
|---|---|---|
| SMoE | 44.73 GB | 0.187 |
| XMoE | 44.76 GB | 0.204 |
| $\sigma$-MoE | 44.79 GB | 0.192 |
| SharedE-V2 | 43.52 GB | 0.182 |
| SharedE-V3 | 43.51 GB | 0.186 |
| TC-MoE | 44.76 GB | 0.201 |
| MoE++ | 42.76 GB | 0.183 |

Two practical patterns are worth highlighting. First, the shared-expert variants occupy a particularly favorable efficiency region. Relative to the standard SMoE baseline, SHAREDE-V2 and SHAREDE-V3 reduce peak training memory by about 1.2 GB while also delivering equal or lower inference latency, and SHAREDE-V2 achieves the lowest latency overall (0.182 s/sample). This is consistent with the architectural intuition discussed above: because some capacity is moved to always-on shared experts, the router only scores the remaining routed experts, which lowers routing overhead in the VLM setting.

Second, Table 15 shows that methods with broadly similar benchmark quality can nevertheless differ meaningfully in system cost. In this setting, inference latency ranges from 0.182 to 0.204 s/sample, even though the main benchmark tables show only modest quality differences across methods. The contrast is especially clear for XMoE and TC-MoE, which are among the slowest methods at inference while remaining close to the rest of the benchmark in peak memory usage. By contrast, MoE++ achieves the lowest peak memory usage, indicating that different architectural modifications can shift different parts of the quality–efficiency trade-off. Taken together, these results reinforce our main practical claim: SMoE methods should be compared not only by end-task quality, but by the full quality–efficiency frontier they induce.

## K Training Benchmark Curves

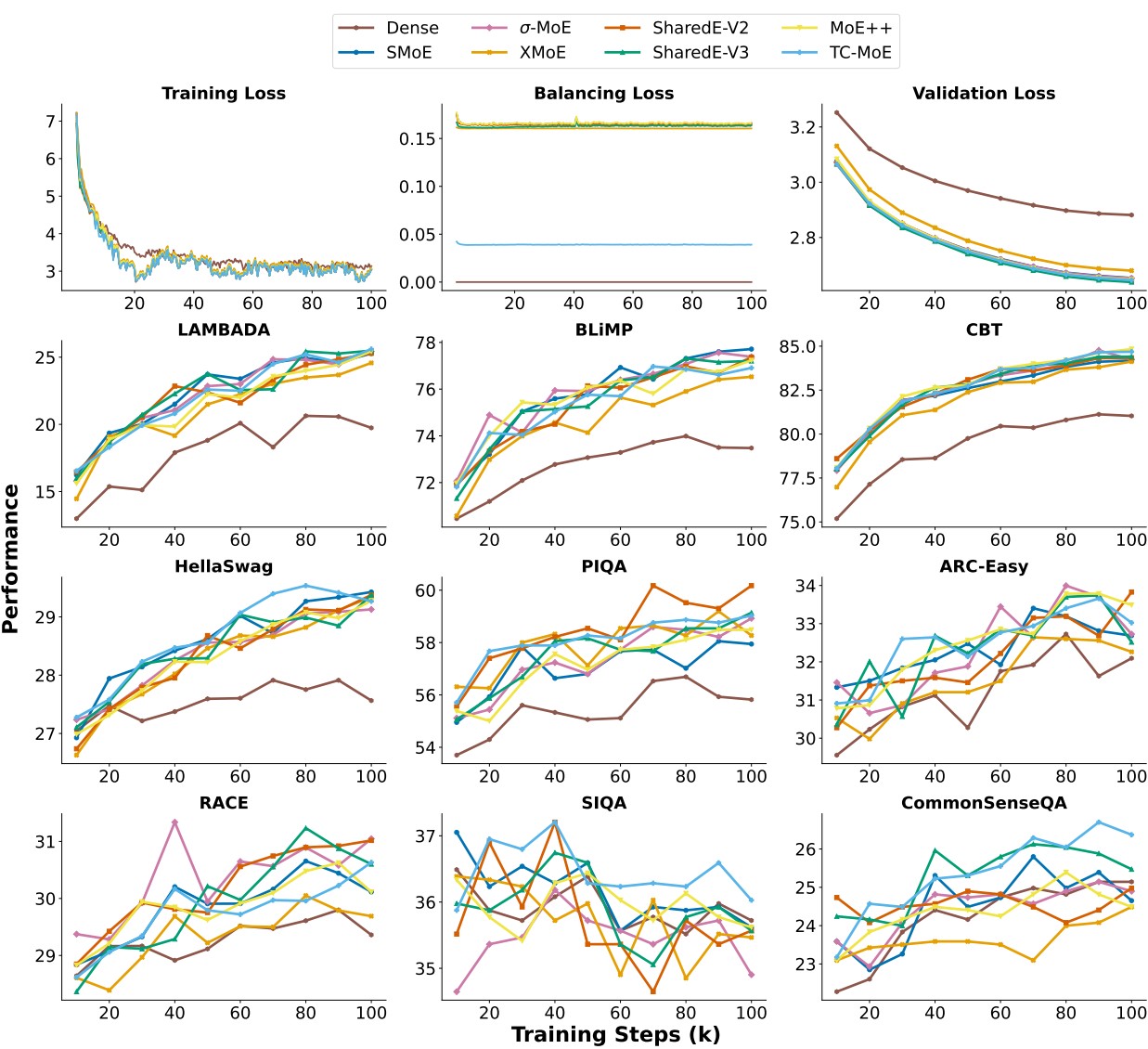

Figure 18: Benchmark curves during training in language modeling tasks for models with 0.15B parameters.

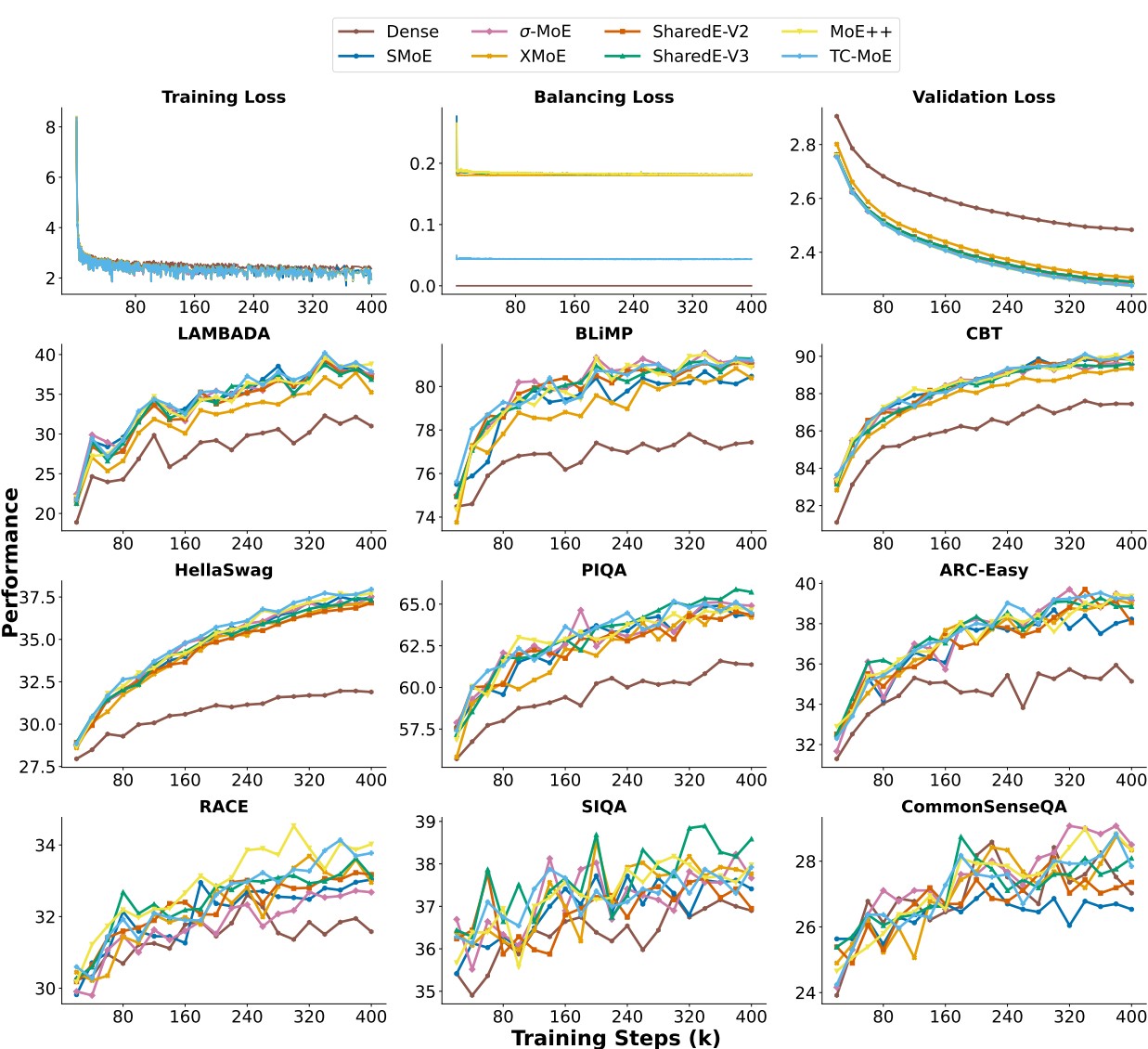

Figure 19: Benchmark curves during training in language modeling tasks for models with 0.68B parameters.

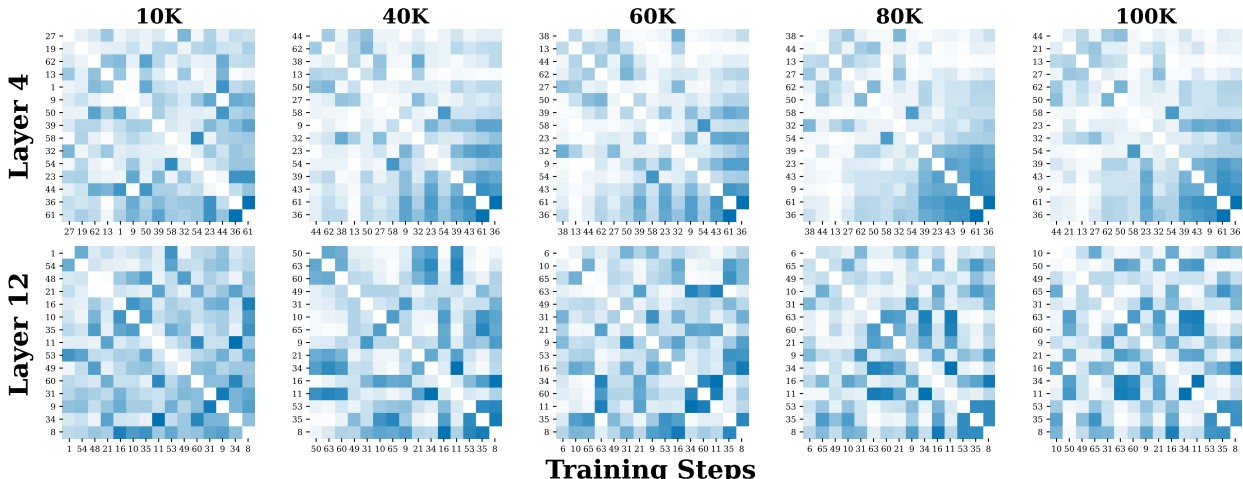

Figure 20: Expert Co-Activation across training XMoE model in a language modeling task.

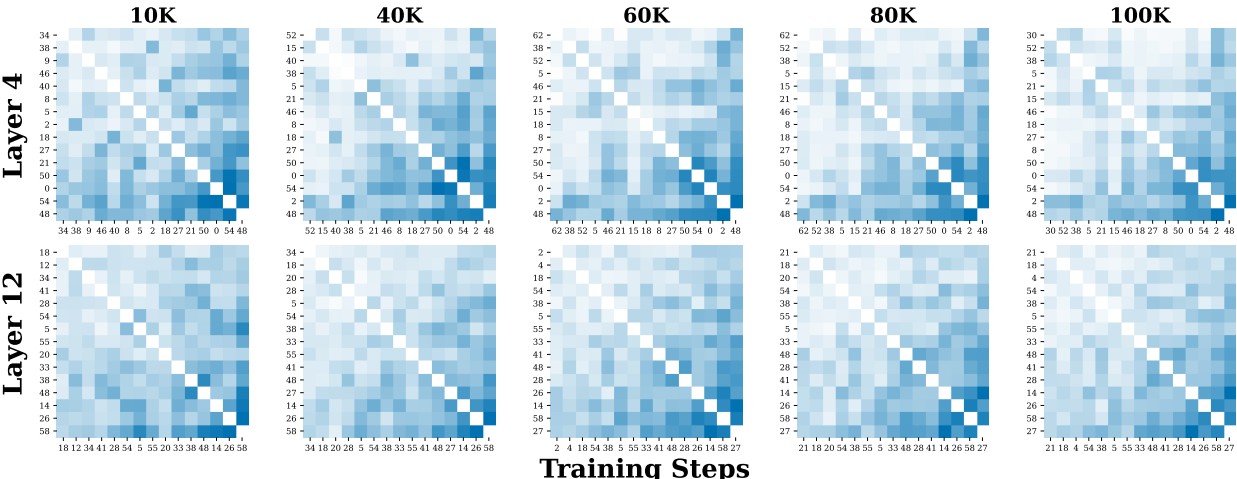

Figure 21: Expert Co-Activation across training $\sigma$-MoE model in a language modeling task.

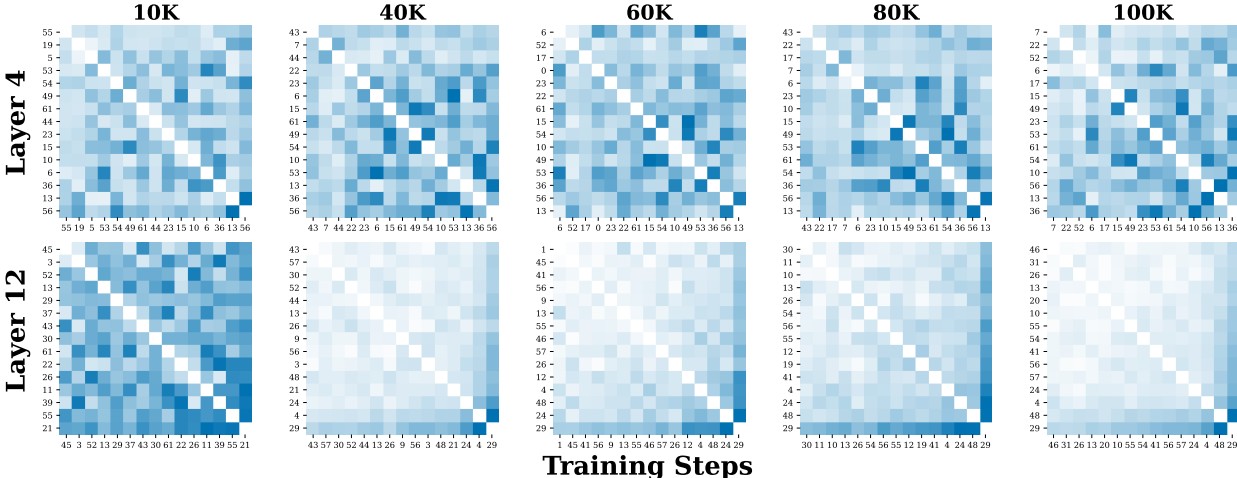

Figure 22: Expert Co-Activation across training SharedE-V2 model in a language modeling task.

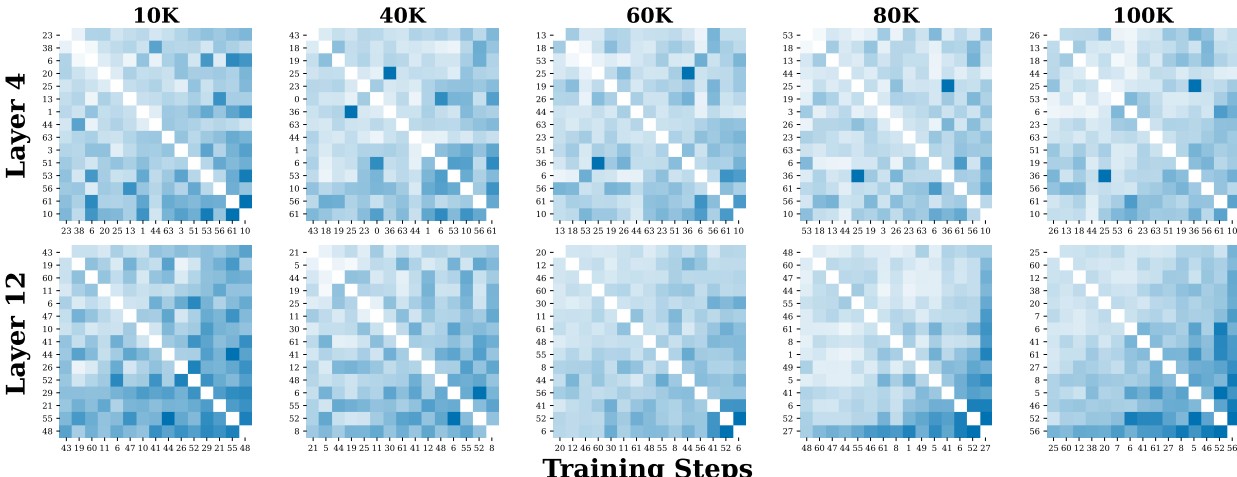

Figure 23: Expert Co-Activation across training SharedE-V3 model in a language modeling task.

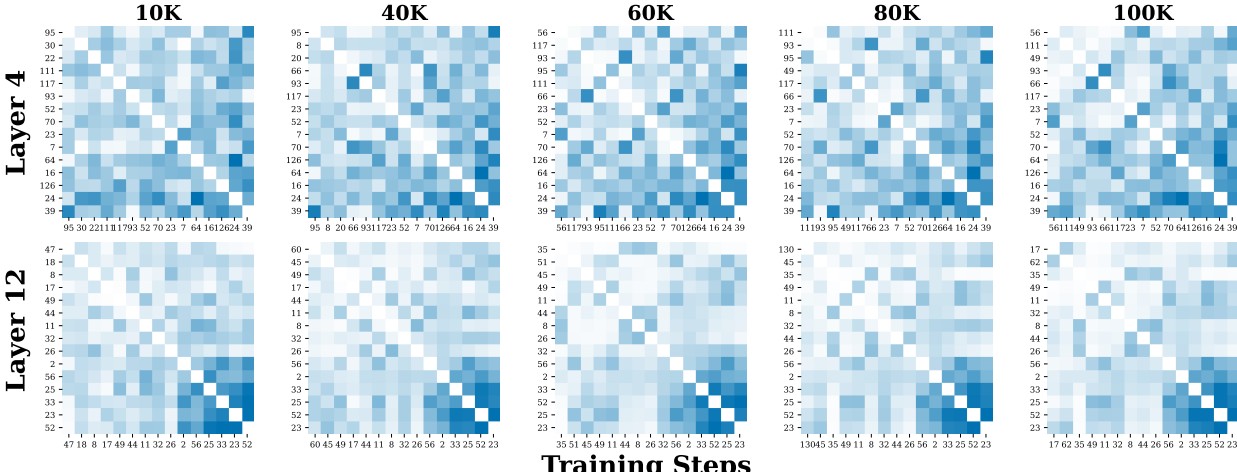

Figure 24: Expert Co-Activation across training TC-MoE model in a language modeling task.

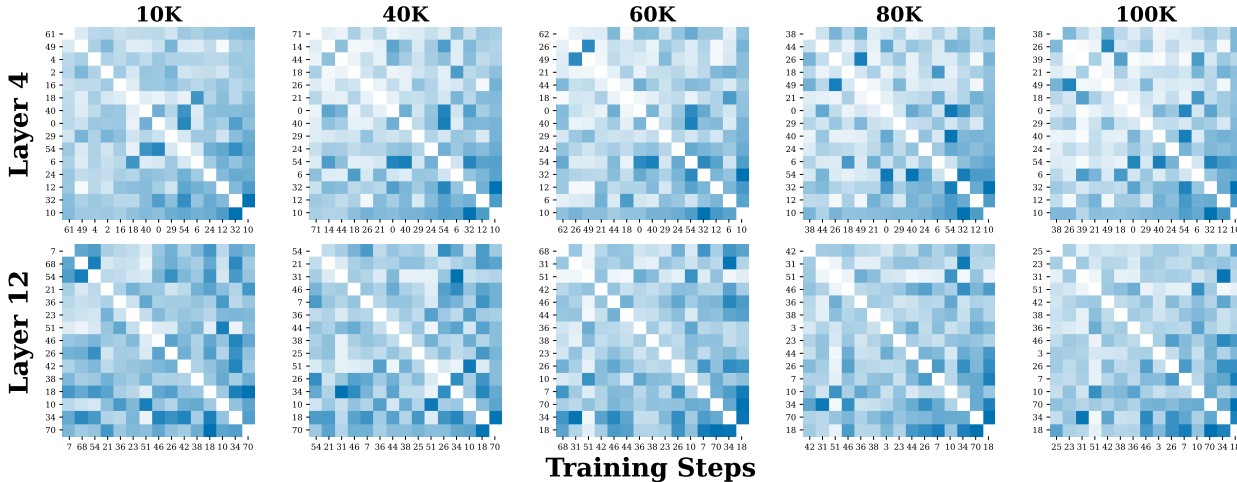

Figure 25: Expert Co-Activation across training MoE++ model in a language modeling task.

