# OpenReview forum: "LibMoE: A Library for Comprehensive Research on Mixture of Experts in Large Language Models"
_TMLR — Accepted by TMLR_

### Review · Reviewer_Czqe · 2026-03-11

**Summary Of Contributions:**

This paper presents LibMoE, a unified and extensible framework designed to facilitate reproducible research on Mixture-of-Experts (MoE) architectures for LLMs and VLMs. Previous research on MoE architectures has typically required substantial computational resources and has been fragmented in the methodologies tested. LibMoE solves these issues by providing standardized implementations of multiple MoE algorithms, modular training pipelines supporting both pretraining and sparse upcycling, and analytical tools for examining expert routing behavior. The framework enables systematic experiments across language and multimodal tasks using relatively modest resources, making large-scale MoE experimentation more accessible to researchers.

Using LibMoE, the authors conduct an extensive empirical study comparing several modern MoE variants and analyzing their internal routing dynamics. Their results show that while different MoE algorithms achieve similar overall task performance, they exhibit meaningful differences in routing stability, expert specialization, and load balancing. The study highlights that routing decisions evolve during training, deeper layers exhibit stronger expert dominance, and initialization choices can significantly affect expert utilization without changing the architecture. Overall, LibMoE provides a practical research platform for MoE-based models.

**Audience:**

Yes

**Audience Explanation:**

I believe the researchers exploring new MoE architectures would be interested in this paper. Beyond reporting benchmark results, the paper provides several analyses that are directly useful for designing future MoE systems. For example, its study on routing stability shows that different MoE variants can have very different expert-change dynamics during training, and that these dynamics differ substantially between from-scratch pretraining and sparse upcycling. This is interesting for researchers proposing new architectures because it suggests that a method that looks stable in one training regime may behave quite differently in another.

**Broader Impact Concerns:**

There are no broader impact concerns.

**Claims And Evidence:**

Yes

**Claims Explanation:**

LibMoE covers a sufficiently broad range of MoE architectures and training settings to support its claim of being a comprehensive framework, spanning both text-only models and vision-language models. It also includes detailed analyses of routing stability, perturbation sensitivity, entropy-based specialization, router margin, expert similarity, and initialization effects. These experiments provide meaningful insights into MoE behavior and largely support the claims made in the abstract about enabling deeper analysis of MoE architectures.

That said, the framework could be strengthened by covering a wider range of design choices that appear in recent MoE literature. For example, the paper does not provide a systematic comparison of sparsity ratios (i.e., the ratio of active parameters to total parameters) or of hybrid architectures that interleave MoE layers with dense MLP layers, as used in modern models such as DeepSeek-V3. In addition, LibMoE does not evaluate system efficiency for training or inference, which is an important consideration when assessing new MoE architectures and training methods in practice.

Overall, I believe the paper provides accurate, convincing, and clear evidence for its main claims, even if its notion of “comprehensive” could be expanded further.

**Requested Changes:**

I do not see any major things that need to be fixed. The paper would be further improved by including evaluations that are discussed above (sparsity ratio, hybrid models, and system efficiency).

- On page 2, the second paragraph, "we present LibMoE a unified framework" seems to need a punctuation after "LibMoE."
- The paper has several issues in the citation: StableMoE, DeepSeek-V3, ReMoE, "Outrageously Large Neural Networks", Yuan 2.0-M32, gpt-oss-120b & gpt-oss-20b model card, and "Auxiliary-Loss-Free Load Balancing" papers are cited twice in two duplicate entries.

---

> ### Author Response · Authors · 2026-04-24
> **Detailed Response to Reviewer Czqe**
>
> We sincerely thank the reviewer for the detailed and highly positive evaluation of our work, as well as for the constructive and insightful suggestions. We are grateful for the reviewer’s thoughtful assessment and encouragement. Below, we provide additional analyses and clarifications in response to the suggestions, particularly regarding sparsity ratios, hybrid architectures, and system efficiency.
>
> ---
>
> **\[R1\] Punctuation and Duplicate Citation Issues**
> We have corrected these editorial issues in the revised manuscript, including the missing punctuation and the duplicate citation entries.
>
> ---
>
> **\[R2\] Sparsity Ratio**
>
> We thank the reviewer for this insightful suggestion. To address this point, we added a detailed sensitivity analysis of the number of active experts in both vision-language training and language-model pretraining in Appendix B. Specifically, in the VLM setting we study $K \\in {1,2,3}$ with $N=6$, while in the language-model pretraining setting we study $K \\in {2,8,32}$ with $N=66$. The new results show that the effect of the active-expert budget is clearly regime-dependent. In language-model pretraining, smaller or moderate values of $K$ improve more quickly in early training, whereas larger $K$ trains more slowly but can achieve stronger final validation performance over a longer horizon; in our setup, $K=8$ provides a particularly good compromise. In contrast, in the VLM setting, moving from $K=1$ to $K=2$ is beneficial, while the gain from $K=2$ to $K=3$ is comparatively small. These findings make the design insight more concrete: there is no single universally optimal active-expert budget, and the preferred choice depends on the target regime and the desired trade-off between efficiency and final quality.
>
> ---
>
> **\[R3\] System Efficiency**
> We agree that system efficiency is an important practical dimension for evaluating MoE methods. In response to this suggestion, we added additional system-level evaluations, including training time and resource usage in Table 14, as well as peak GPU memory usage and inference latency in Table 15 of Appendix J.
>
> In the VLM setting, these results provide a clearer view of the practical trade-offs across MoE methods. While several standard SMoE variants achieve broadly similar accuracy under matched compute and data budgets, their system-level efficiency differs meaningfully.
>
> In particular, the shared-expert architectures again emerge as strong practical choices: SharedE-V2 and SharedE-V3 require less training time than most alternatives, use less peak GPU memory, and achieve faster inference latency, while remaining competitive in model quality.
>
> We have revised the manuscript accordingly to make this quality--efficiency trade-off more explicit.
>
> ---
>
> **\[R4\] Hybrid Dense–Sparse Architectures.**
> We thank the reviewer for this helpful suggestion. We agree that hybrid dense–MoE layouts are practically important. To address this point, we added a new experiment in Appendix D comparing Hybrid SharedE-V3 with the fully sparse SharedE-V3 model in the 5.67B VLM setting. While the hybrid design is slightly faster to train, it performs worse overall, reducing average accuracy from 50.86 to 50.43 after training on LLaVA-665K.
>
> Importantly, this observation is consistent with prior studies on expert placement. MoE-LLaVA\[1\] and MoExtend\[2\] report that introducing experts only in the second half of the model underperforms applying experts across all layers. These findings align with our result: retaining dense early layers may reduce the benefits of sparse expert routing compared with a fully sparse allocation. Interestingly, this view is also consistent with the latest frontier-scale model, DeepSeek-V4 \[3\], which removes the dense layers used in earlier hybrid designs and replaces them with SMoE layers throughout the architecture (see Section 2.1 of \[3\]). Thus, while hybrid dense–MoE architectures remain an important design direction, our controlled experiment suggests they are not universally superior and should be evaluated under matched settings.
>
> ---
>
> **References:**
> \[1\] Lin, Bin, et al. MoE-LLaVA: Mixture of Experts for Large Vision-Language Models. IEEE Transactions on Multimedia, 2026\.
>
> \[2\] Zhong, Shanshan, et al. MoExtend: Tuning New Experts for Modality and Task Extension. ACL SRW, 2024\.
>
> \[3\] https://huggingface.co/deepseek-ai/DeepSeek-V4-Pro/blob/main/DeepSeek_V4.pdf

---

### Review · Reviewer_xDms · 2026-03-16

**Summary Of Contributions:**

This paper introduces LibMoE, a unified, open-source framework designed to lower the computational barrier for Mixture-of-Experts (MoE) research by enabling systematic experimentation on limited resources (e.g., 4×H100 GPUs). Its core contribution is a comprehensive, standardized benchmark of several state-of-the-art MoE algorithms across language model pretraining and vision-language sparse upcycling.

**Strengthness**

 - The primary contribution of this paper is LibMoE, a modular, open-source library designed to lower the barrier to entry for Mixture-of-Experts (MoE) research. It is specifically engineered to enable meaningful experiments under realistic constraints, making MoE research more accessible to the wider community.

 -  This paper provides a systematic comparison of seven state-of-the-art MoE algorithms—SMoE, XMoE, σ-MoE, SharedE-V2/V3, TC-MoE, and MoE++—across multiple regimes. It introduces and applies transparent analytical tools to probe routing dynamics in a way rarely seen in prior work. This analysis provides several novel and actionable insights, including those regarding routing stability, routing optimality, task specialization, and initialization effects.

**Limitations**

 - While the smaller scale of the models enables a larger community to research MoE, it also raises a significant concern: would the conclusions hold for product-grade models, which typically use trillions of tokens and hundreds of billions of parameters? As the authors acknowledge, some MoE behaviors may only fully emerge at those extreme scales. Therefore, a consistency analysis between small-scale and large-scale models would be beneficial.

 - In contrast to the abundance of perspectives used to analyze certain aspects, such as routing dynamics, a major problem is identifying the key takeaway message. The paper offers many insights, noting that in some perspectives and for some tasks, method A is better than method B. However, it is difficult to extract insights that would be helpful in selecting MoE methods or devising a new one. Furthermore, readers who are not entirely familiar with the existing seven MoE methods might find the paper difficult to follow.

**Audience:**

Yes

**Audience Explanation:**

Yes. This paper would interest a significant portion of the TMLR audience.
- Practitioners with limited resources gain a validated toolkit (LibMoE) and clear roadmap for conducting MoE research on accessible hardware (e.g., 4×H100s), directly addressing a major barrier in the field.
- Researchers focused on model analysis receive novel insights into routing dynamics (stability, specialization, initialization effects) that go beyond surface-level benchmark comparisons.
- The community seeking standardization benefits from a unified framework and comprehensive benchmark that establishes a common ground for fair, reproducible MoE comparisons.

**Claims And Evidence:**

Yes

**Claims Explanation:**

The evidence in the submission is generally accurate and convincing .

- The paper provides clear, quantitative support for LibMoE's accessibility (detailing exact compute resources and training times) and for the marginal performance differences among algorithms
- Strong behavioral analyses are provided, where well-designed experiments and clear figures (e.g., on routing stability, optimality, and specialization) directly support each analytical insight.
- Claims about broader impact are reasonable extrapolations, but are supported by more indirect evidence, as the experiments, while rigorous, remain at an academic scale.

**Requested Changes:**

- Demonstrate the consistency of conclusions as data and parameter scales increase.
- Further clarify the key insights regarding the design of Mixture of Experts (MoE) frameworks.

---

> ### Author Response · Authors · 2026-04-24
> **Detailed Response to Reviewer xDms**
>
> We sincerely thank the reviewer for the thoughtful and positive evaluation of our work, especially for recognizing LibMoE’s contribution as an accessible framework for Mixture-of-Experts research, the value of its standardized benchmark and routing analyses, and its potential to enable systematic MoE experimentation under limited computational resources.  Below, we address the reviewer’s concerns in turn and provide additional analyses and clarifications in response.
>
> ---
>
> **\[R1\] Scale Consistency and Generalization to Product-Grade MoE Models**
> We thank the reviewer for raising this important limitation. We agree that an important open question is whether conclusions drawn from accessible-scale experiments remain valid for product-grade MoE models trained with substantially larger data and compute budgets.
>
> We emphasize that direct and comprehensive validation at trillion-token scales is infeasible for most research labs, including ours. Instead, we focus on a compute-accessible setting representative of the resources available to a broader research community (with our main MoE VLM runs requiring roughly 16–36 hours). Within this scope, to address the reviewer’s generalization concern, we complement our controlled LibMoE experiments with additional analysis of Qwen3-VL-30B-A3B \[1\], a representative large-scale SMoE model trained at trillion-token scale. Specifically, we include Qwen3-VL results in the analyses of Section 5(c)–5(f), covering expert-allocation entropy, expert-weight allocation, router margin, and expert similarity.
>
> Encouragingly, these large-scale observations are qualitatively consistent with our main findings. In particular, Qwen3-VL exhibits broad routing characteristics similar to those identified in LibMoE, suggesting that several of our behavioral conclusions are not merely artifacts of small-scale experimentation.
>
> At the same time, the main goal of LibMoE is to provide a practical framework that enables researchers with limited computational resources to conduct meaningful MoE experiments under controlled settings. For this reason, we prioritize accessible and reproducible benchmarking rather than scaling the study to much larger training budgets. We therefore present the Qwen3-VL comparison as supportive large-scale evidence that several key routing patterns observed in LibMoE remain relevant in a frontier-scale model.
>
> ---
>
> **\[R2\] Further clarify the key insights regarding the design of Mixture-of-Experts (MoE) methods.**
>
> We thank the reviewer for this helpful suggestion. We agree that the original manuscript could more clearly articulate the key design insights for Mixture-of-Experts (MoE) methods.
>
> In response, we substantially revised Section 6, Summary of Key Results, to clarify both the high-level empirical conclusion and the resulting design principles. In particular, we now explicitly state that, once compute and data budgets are matched, performance differences among current SMoE methods are generally modest across both vision--language sparse upcycling and language-model pretraining, and no single method emerges as a universal winner across regimes.
>
> We believe this clarification is important because it sharpens the main message of the paper: the contribution of LibMoE is not only benchmark comparison, but a principled understanding of how MoE methods differ in routing stability, specialization, robustness, efficiency, and optimization behavior. We also make the practical takeaway more explicit by highlighting that SharedE-V2 and SharedE-V3 stand out as strong practical choices due to their competitive quality together with favorable memory, training-efficiency, and inference trade-offs.
>
> Building on this framing, we reorganized the discussion around three clearer design principles: (1) prioritize methods that remain strong across training regimes, while using regime-specific differences diagnostically; (2) do not equate sharper routing with better routing, since specialization and robustness trade off; and (3) prefer lightweight interventions, such as router initialization choices, before introducing additional architectural complexity.
>
> We also revised the concluding paragraph to make the evaluation criteria for future MoE methods more explicit: a strong method should remain competitive across regimes under matched compute, remain robust under routing perturbations, and demonstrate better specialization or capacity utilization rather than merely sharper routing.
>
> We hope these revisions make the key design insights of the paper substantially clearer and more actionable.
>
> ---
>
> **\[R3\] Clarifying the Benchmarked SMoE Variants**
>
> We thank the reviewer for this suggestion. To address this, we have added Appendix A, Overview of Benchmarked SMoE Algorithms, which summarizes each method’s routing rule, structural distinctions, and design motivation in short descriptions.
>
> ---
>
> References:
>
> \[1\] https://huggingface.co/Qwen/Qwen3-VL-30B-A3B-Instruct

---

> > ### Author Response · Authors · 2026-05-05
> > **Appreciation and Follow-up on Discussion**
> >
> > Dear Reviewer **xDms**,
> >
> > We have posted a detailed response addressing your concerns and clarifying the revisions we have made. As the discussion deadline is approaching, we would be very grateful if you could kindly take a look at our response and join the discussion at your convenience. We would also sincerely appreciate any further questions, comments, or suggestions you may have.
> >
> > Best regards,
> >
> > The Authors

---

### Review · Reviewer_pDnB · 2026-04-10

**Summary Of Contributions:**

This paper introduces LibMoE, a unified framework for studying sparse Mixture of experts models. The contribution is primarily systematic and empirical rather than algorithmic. Specifically, the paper provides a common training and evaluation benchmark spanning both language-model pretraining and vision-language sparse upcycling, integrates seven recent SMoE variants under standardized settings, and adds analysis tools for inspecting routing behavior, specialization, expert diversity, load balancing, and expert utilization. On top of the framework, the paper conducts a fairly broad empirical comparison.

**Key strengths**
- The practical value of the paper is clear: the authors reduce the cost of running SMoE experiments, make comparisons more standardized and reproducible, and go beyond the empirical performance by examining the detailed effects.
- The controlled comparisons suggest that, under a unified setup, many existing SMoE variants deliver only incremental improvements over vanilla SMoE, which is an important result for the community.

**Audience:**

Yes

**Audience Explanation:**

MoE has become an important component of modern LLM research and deployment, so I believe part of the TMLR audience would be interested in the findings of this paper. This paper is also useful in a more practical sense: it introduces a codebase for more standardized and comparatively fair evaluation of SMoE methods, and it accompanies that release with a fairly extensive empirical study. In addition to the benchmark results, the paper includes informative analyses of routing behavior and training-regime differences. Even for readers who may not use LibMoE directly, the evaluation setup, analysis tools, and empirical findings are likely to be of interest.

**Broader Impact Concerns:**

I do not have major broader impact concerns.

**Claims And Evidence:**

Yes

**Claims Explanation:**

For the paper’s main claims, the evidence is generally clear, and in most cases reasonably supportive. Tables 1 and 2 are broadly consistent with the paper’s central observation that recent MoE variants tend to remain fairly close in end-task performance under the experimental setting considered here. The paper also includes a fairly detailed analysis of routing behavior, covering expert selection dynamics, routing stability, expert specialization, and load balancing, which adds useful depth beyond aggregate benchmark scores. In addition, the distinction between full pretraining and sparse upcycling is a worthwhile part of the paper and helps contextualize some of the behavioral differences reported in the analysis.

**Requested Changes:**

- Please report peak training memory usage for each method, as this would strengthen the paper’s practical comparison.
- Please discuss sensitivity to the number of active experts in more detail because the current configurations are still somewhat limited.
- Please report multi-seed mean±std results for the main tables, as this would clarify whether the small gains are statistically meaningful.

---

> ### Author Response · Authors · 2026-04-24
> **Detailed Response to Reviewer pDnB**
>
> We sincerely thank the reviewer for the thoughtful and positive evaluation of our work, especially for recognizing the practical value of LibMoE, the clarity of the evidence, and the importance of standardized and reproducible evaluation for MoE research. Below, we address the reviewer’s concerns in detail and provide additional clarifications.
>
> ---
>
> **\[R1\] Peak Training Memory Usage Across MoE Methods**
>
> Thank you for this valuable suggestion. In response, we added a new analysis in Appendix J.2 reporting the peak training memory usage of all seven MoE methods, together with per-sample inference latency. These results strengthen the practical conclusions of the paper by showing that methods with similar benchmark quality can nevertheless differ substantially in system efficiency. In particular, the shared-expert variants again stand out as strong practical choices: SharedE-V2 and SharedE-V3 achieve lower peak memory usage than most alternatives and also provide lower inference latency, while remaining competitive in model quality. This further supports our main claim that MoE methods should be evaluated not only by final accuracy, but also by their quality-efficiency trade-offs.
>
> ---
>
> **\[R2\] Sensitivity to the Number of Active Experts**
>
> We thank the reviewer for this insightful suggestion. To address this point, we added a detailed sensitivity analysis of the number of active experts in both vision-language training and language-model pretraining in Appendix B. Specifically, in the VLM setting we study $K \\in {1,2,3}$ with $N=6$, while in the language-model pretraining setting we study $K \\in {2,8,32}$ with $N=66$. The new results show that the effect of the active-expert budget is clearly regime-dependent. In language-model pretraining, smaller or moderate values of $K$ improve more quickly in early training, whereas larger $K$ trains more slowly but can achieve stronger final validation performance over a longer horizon; in our setup, $K=8$ provides a particularly good compromise. In contrast, in the VLM setting, moving from $K=1$ to $K=2$ is beneficial, while the gain from $K=2$ to $K=3$ is comparatively small. These findings make the design insight more concrete: there is no single universally optimal active-expert budget, and the preferred choice depends on the target regime and the desired trade-off between efficiency and final quality.
>
> ---
>
> **\[R3\] Multi-Seed Mean$\\pm$Std Results for the Main Tables**
>
> Thank you for raising this important point. We added multi-seed mean$\\pm$std results in Appendix C for both the vision-language and language-model settings, using three random seeds (42, 128, and 456). These additional results further strengthen the robustness of our conclusions. First, they confirm that the performance gaps among current SMoE methods remain narrow under matched budgets, indicating that our main findings are not artifacts of a single run. Second, they show that the relative ranking of methods is benchmark- and regime-dependent, rather than dominated by one universal winner. In the language-model setting, several methods form a closely matched leading group, while in the VLM setting the differences are also small and the best method depends on the benchmark.

---

### Author Response · Authors · 2026-05-07
**General Response and Summary of Revisions**

Dear Action Editor and Reviewers,

We sincerely thank the Action Editor and all reviewers for their careful reading, constructive feedback, and thoughtful evaluation of our work. We are especially grateful for the reviewers’ recognition of LibMoE’s practical value, accessibility, standardized benchmarking, and detailed routing analysis. We deeply appreciate the constructive suggestions, which have helped us substantially improve the clarity, rigor, and overall presentation of the manuscript.

---

### **Key Strengths Highlighted by Reviewers**


-  **Insightful routing and expert-dynamics analysis.** Reviewers consistently appreciated that LibMoE goes beyond reporting aggregate benchmark performance and instead provides deeper analyses of MoE routing behavior and expert dynamics. Reviewer pDnB emphasized that the paper complements its benchmark results with useful analytical tools and empirical studies, making the evaluation framework and findings broadly valuable to the community. Reviewer xDms further highlighted that LibMoE offers novel insights into routing dynamics including routing stability, expert specialization, and initialization effects rather than relying solely on surface-level performance comparisons. Reviewer Czqe similarly noted the detailed investigations of routing robustness, entropy-based specialization, router margin, and expert similarity, and emphasized that the paper’s analyses beyond benchmark results are directly useful for designing future MoE systems.

- **Strong empirical support and community relevance.** Reviewer pDnB found the paper’s claims generally well supported by clear and reasonably supportive evidence, while Reviewers xDms and Czqe similarly considered the empirical analyses accurate, convincing, and informative. More broadly, reviewers indicated that LibMoE would be valuable to the TMLR community, particularly for researchers and practitioners seeking accessible, reproducible, and standardized evaluation of modern MoE systems.

- **Practical contribution and accessibility.** Reviewers consistently recognized LibMoE as a valuable and practical contribution to the MoE community. Reviewer pDnB highlighted its ability to reduce the cost of SMoE experimentation while improving reproducibility and standardization. Reviewer xDms emphasized that LibMoE lowers the barrier to entry for MoE research under realistic academic-scale resources, and Reviewer Czqe appreciated the framework’s unified and extensible design for systematic experimentation.

- **Standardized benchmarking and broad evaluation.** Reviewer pDnB appreciated the unified benchmark spanning both language-model pretraining and vision-language sparse upcycling. Reviewer xDms highlighted the systematic comparison of seven representative MoE methods across multiple regimes, while Reviewer Czqe noted that LibMoE covers a broad range of architectures and training settings, making the framework comprehensive and practically useful.

---

### **Major Reviewer Concerns and Our Response**

- **Scale consistency and relevance to larger MoE models.** While most reviewer requests involved additional analyses and clarifications that we have addressed in the revised manuscript, Reviewer xDms raised a broader question about whether observations from accessible-scale experiments remain informative for larger product-scale MoE systems trained with substantially greater compute and data. We appreciate the reviewer for highlighting this concern. To directly address it within a feasible compute budget, we added Qwen3-VL-30B-A3B as a representative large-scale SMoE reference model in our routing-behavior analyses. Specifically, we extended the analyses of expert-allocation entropy, expert-weight allocation, router margin, and expert similarity to include Qwen3-VL. This addition provides a concrete large-scale reference point for assessing whether the qualitative routing patterns observed in LibMoE also appear in a substantially larger model. Encouragingly, several of these patterns remain consistent in the Qwen3-VL analysis, providing supporting evidence that the behaviors studied by LibMoE are not merely artifacts of the small-scale settings used in our main experiments. While we do not claim that this analysis replaces full frontier-scale validation, it substantially strengthens the paper’s response to the reviewer’s concern and reinforces LibMoE’s central goal of enabling reproducible, transparent, and academically accessible MoE research while maintaining a meaningful connection to modern large-scale MoE models.

---

> ### Author Response · Authors · 2026-05-07
> **Summary**
>
> In summary, we sincerely thank Reviewer pDnB, Reviewer xDms, and Reviewer Czqe for their constructive feedback and thoughtful evaluation, and we thank the Action Editor for overseeing the review process. We have carefully addressed all reviewer concerns and incorporated the requested analyses, clarifications, and editorial corrections into the revised manuscript. These revisions have substantially improved the clarity, rigor, and overall quality of the paper. We are grateful for the reviewers’ time and expertise, and we respectfully hope that the revised version fully addresses the concerns raised during the review process.
>
> Best regards,
>
> The Authors

---

### Decision · Action_Editor_JGvb · 2026-05-19

**Recommendation:** Accept as is

**Additional Comments:**

Camera-ready reminders:
1. Figure 8 is missing its header.
2. Ensure the LibMoE code repository URL is included and accessible (non-anonymized).
3. Confirm Appendix A's overview of the seven SMoE variants is complete and self-contained for readers unfamiliar with all methods.

**Audience:**

Yes

**Audience Explanation:**

All three reviewers answer Yes. AE agrees that the case is strong:

1. MoE is a major architecture in current frontier LLMs and the cost of methodological MoE research is a real bottleneck. A unified library that enables meaningful experiments at 4×H100 directly addresses that bottleneck.
2. The integration of seven SMoE variants (SMoE, σ-MoE, XMoE, SharedE-V2/V3, MoE++, TC-MoE) under a single training/evaluation harness, with extension of LMMS-Eval for VLMs, is an independently useful infrastructure.
3. The analytical toolkit (ECR, EAE, EWA, router-margin profiles, perturbation-based routing-optimality tests, expert-similarity diagnostics) is a reusable metric beyond this paper.
4. The "negative" headline result, that modern SMoE variants rarely beat vanilla SMoE under matched budgets, is the kind of finding TMLR exists to publish: rigorously evidenced, useful to the community, and unlikely to find a home at a venue that prioritizes positive deltas.

**Claims And Evidence:**

Yes

**Claims Explanation:**

Yes. All three reviewers answered Yes on this criterion. The revised manuscript improved in multiple ways:

1. The core empirical claim, under matched compute and data budgets, contemporary SMoE variants show only modest end-task differences, with no clear winner, is well supported by Tables 1–2 and reinforced by three-seed mean+-std results in Appendix C.
2. The mechanistic claims about routing dynamics are supported by the analytical metrics introduced in Section 5 and partially corroborated against the external Qwen3-VL-30B-A3B reference. The Qwen3-VL anchoring is qualitative rather than quantitative, but authors reasonably frame it as "supporting evidence" that observed patterns are not artifacts of small-scale settings.
3. The framework-level claims (accessible 4×H100 compute, training-time accounting, peak memory, inference latency) are now quantitatively documented in Appendix J (Tables 14 and 15).
4. The lightweight-initialization claim (router std=0.02 vs. 0.04/0.06 improving load balance) is a clean, controlled result.

---

> ### Author Response · Authors · 2026-06-22
> **Camera-Ready Changes Implemented**
>
> Dear Action Editor,
>
> Thank you very much for your recommendation to accept our paper and for your careful handling of our submission.
>
> In the camera-ready version, we have addressed all the comments and suggestions in your comments. Specifically:
>
> * We added the missing header for Figure 8.
> * We ensured that Appendix A provides a complete and self-contained overview of all seven SMoE variants.
> * The LibMoE code repository has been de-anonymized and is now publicly accessible.
>
> We sincerely appreciate your time, effort, and support throughout the review process.
>
> Best regards,
>
> The Authors